

# Naturally fractured reservoir characterization in heterogeneous sandstones: insight for Uranium In Situ Recovery (Imouraren, Niger)

Maxime Jamet [1,*], Gregory Ballas [1], Roger Soliva [1], Olivier Gerbeaud [2], Thierry Lefebvre [2], Christine Leredde [1], Didier Loggia [1]

[1] Geosciences Montpellier, UMR 5243, Université de Montpellier, Place Eugène Bataillon, 34095 Montpellier Cedex 5, France
[2] ORANO, BU Mines, Direction Géosciences, 125 avenue de Paris, F-92320 Châtillon, France
[*] Corresponding author.

*Correspondence to*: (Maxime Jamet) maxime.jamet@umontpellier.fr

**Abstract.** This study delves into the characterization of a complex reservoir, the Tchirezrine II sandstone unit in North Niger, crucial for potential Uranium In Situ Recovery (ISR) in a naturally fractured and faulted context. Employing a multifaceted approach, including well log data, optical borehole imagery, and hydrogeological tests, alongside satellite-based lineament analysis, this study provides a comprehensive understanding of the structures and its impact on fluid flow. Lineament analysis reveals scale-dependent patterns, consistent with spatially homogeneous joint networks restricted to mechanical units, as well as nearly scale-invariant patterns, better corresponding to spatially heterogeneous fault networks. Various deformation structures are detected from borehole imagery, including Mode I fractures, cataclastic deformation bands, and brecciated-cataclastic fault cores. The Tchirezrine II reservoir displays heterogeneous porosity and permeability related to its fluviatile sedimentary context. These data differ from traditional porosity-permeability relationship obtained in sandstone reservoir matrix but are instead consistent with Nelson's classification, emphasizing the impact of deformation structures on such petrophysical properties. Hydrological tests have been implemented into a zone of E-W trending deformation structures, revealing a strong permeability anisotropy of this heterogeneity. This strong E-W anisotropy is consistent with the presence of the observed E-W structures, i.e. with a drain behaviour of Mode I open fractures and a sealing behaviour of both cataclastic bands and fault rocks. Considering implications for ISR mining, this study allows discussing the interplay between fractures, faults, and fluid flow properties. It suggests that a well pattern perpendicular to the main permeability orientation can attenuate channelled flow, thus improving contact of the leach solution with the mineralized matrix. These results provide an integrated approach and multi-scale characterization of Naturally Fractured Reservoir (NFR) properties in sandstone, offering a basis for optimization of NFR production such as ISR development.

**Key words:** Naturally Fractured Reservoir, Cataclastic deformation bands, Fault seal, Lineament analysis, Porosity-Permeability relationship, Uranium ore, Sandstone reservoir

## 1 Introduction

Geological reservoirs have long played a crucial role in a variety of fields, from water resources management (Keller et al., 2000) to oil and gas extraction (Goodwin et al., 2014). Today, their importance extends to addressing environmental concerns, especially the transition to low-carbon energies (Evans et al., 2009), encompassing activities such as $CO_2$ sequestration (Qi et al., 2023), hydrogen storage and production (Sambo et al., 2022), geothermal energy (Moeck, 2014), nuclear waste repository (Rempe, 2007) and In Situ Recovery (ISR) of metallic resources (Seredkin et al., 2016). Geological reservoirs are complex media where petrophysical properties (i.e. mainly porosity and permeability) control the capacity to contain and transport the fluids (Bear, 1972). These properties are affected by numerous and imbricated sedimentary-diagenetic and tectonic processes that makes the evaluation of reservoir quality a recurring challenge. This complexity is particularly evident in fractured and faulted environments, known as Naturally Fractured Reservoirs (NFRs) (Narr et al., 2006; Nelson, 2001). The NFRs are



classified by Nelson (2001) following the relative contribution of the fractures and the matrix to the porosity and permeability values. This classification was applied in numerous NFR characterizations (see Allan and Sun, 2003, for a review), especially in tight matrix reservoirs (Evans and Lekia, 1990; Harstad et al., 1995; Lee and Hopkins, 1994; Northrop and Frohne, 1990;

Olson et al., 2009). However, the application of such classification remains a challenge in high porosity matrix reservoirs and/or polyphased tectonic framework, where various processes can make the deformation structures more or less favourable or penalizing for petrophysical properties (Antonellini and Aydin, 1994; Aydin and Johnson, 1978; Ballas et al., 2015; Fossen et al., 2017; Wilkins et al., 2020). Structural and petrophysical properties of NFRs remain poorly constrained in reservoir composed of both tight and porous sedimentary bodies (e.g. Doyle and Sweet, 1995). In such a context, it is essential to study

the arrangement of deformation structures and quantify their respective petrophysical impact within the different parts of the reservoir (vertical and lateral variations in mechano-stratigraphic properties). A better understanding of the relationships between the matrix characteristics, the deformation arrangement and processes, and the petrophysical properties is fundamental to improve reservoir management in such heterogeneous geological systems (Sonntag et al., 2014). This issue is of first importance for potential ISR exploitation of metallic ore deposits, such as Uranium, in mixed matrix - NFR context.

Fluvial sandstone sequences are highly heterogeneous reservoirs, providing a real challenge to analyse the great diversity of structures they can contain and their role on fluid flow. Sedimentological variations within sandstone bodies, encompassing both vertical and lateral dimensions, introduce complexities arising from porosity, grain size, sorting, shape variations, and mineral content (Gibling, 2006; Miall, 1988; Morad et al., 2010). These factors introduce large heterogeneity of petrophysical properties within reservoir units. These factors also significantly shape deformation mechanisms occurring in sandstone

reservoirs (Aydin et al., 2006; Fossen et al., 2007), following the brittle-ductile transition of such porous rocks (Wong et al., 1997; Wong and Baud, 2012). Mode I fracture or disaggregation structures are preferentially formed in low-porosity, fine-grained and poorly-sorted sandstones whereas compactional-shear deformation bands, with cataclastic behaviour, are favoured in highly-porous, coarse-grained and well-sorted ones (Ballas et al., 2015; Fossen et al., 2017; Schultz et al., 2010). The shape and the composition of grains, especially the clay content (Antonellini et al., 1994; Fisher and Knipe, 2001; Gibson, 1998),

secondly contribute to the initiation of various deformation structures. A precise characterization of the host sandstone properties is then necessary to understand the typology of deformation mechanisms and evaluate their impact on the reservoir properties in such context. Small-scale structures such as previously mentioned are generally observed in borehole from NFR and are potentially related to large-scale structures such as fault systems. Conversely, small-scale structures can be totally independent to large-scale faults, such as joint sets and deformation band networks (e.g. Pollard and Aydin, 1988; Soliva et al.,

2016; Mayolle et al., 2019). This diversity of structure and spatial organisation implies multi-scale transfer properties in NFR that are still poorly described and understood, especially for their impact on fluid flow (Warren and Root, 1963; Nelson, 2001). It is therefore needed to provide examples of the characterisation of the role of such structures on fluid flow in sandstone NFR, which actually requires multi-scale and multi-method investigation.

In this paper, we use a multifaceted approach to constrain the properties of a reservoir in a context of heterogeneous fluviatile-
sandstone and polyphased tectonic area. Our study is based on (1) outcropping areas of the Tchirezerine II reservoir (North Niger) to study deformation patterns by satellite-based analysis and statistical interpretations of lineaments, (2) well log data, including optical borehole imagery, geophysical data (Sonic porosity and Nuclear Magnetic Resonance (NMR) permeability), and (3) hydrogeological tests. The Tchirezrine II reservoir, at the Imouraren site, is presently studied for potential Uranium ISR production, never completed in such a complex NFR context. Understanding fluid flow in such NFR is crucial to optimize

ISR production cells, where channelized flow, carried by deformation structures, provides potential bypass for leaching solution, limiting its access to the Uranium ore. The combined approach proposed here aims to provide an integrated comprehensive characterization of complex NFR, encompassing both subsurface and surface data. We interpret and discuss



our findings in terms of spatial organization and petrophysical properties of the structural network, and of the resulting anisotropy of permeability driven by these structures.

## 2 Geological setting

### 2.1 Tectonic framework

The Tim Mersoï basin (Niger) is the northern part of the Iullemeden basin that covers a large part of the West Niger in Central North Africa (Figure 1a). It is located at the intersection of the West African Craton and the East Saharan Craton and forms a north-south trench resting on the Adrar des Iforas Massif to the west, the Hoggar Massif to the north and the Aïr Massif to the east. The basin has experienced a polyphased tectonic evolution, with extensional phases during the Visean, the Lower Permian and the Atlantic rifting (Sempere and Beaudoin, 1984; Valsardieu, 1971), and compressional phases during Hercynian orogeny and Late Cretaceous Alpine event (Gerbeaud, 2006; Guiraud et al., 1981; Yahaya and Lang, 2000). This complex tectonic evolution leads to the formation of various deformation structures affecting both the crystalline basement and the Devonian to Lower Cretaceous sediment infilling (Figure 1b) (Gerbeaud, 2006; Guiraud et al., 1981; Valsardieu, 1971; Yahaya, 1992). These structures include N-S, N30°, N70-80°, and N140-150° fault systems (Gerbeaud, 2006; Valsardieu, 1971). This structural framework was reworked during the late Cretaceous compressional event: the N30° faults in reverse motion, the dextral strike-slip motion of the N070°E faults and the sinistral strike-slip reactivation of the N-S Arlit fault. The Imouraren U. deposit is located in a syncline structure that borders the Arlit fault to the east (Figures 1b and c)). The Madaouela N030°E fault to the North and the Magagi N070°E fault to the South delimit the borders of the Imouraren site. Numerous small-scale faults and fractures are interpreted in the sedimentary reservoir hosting the Imouraren deposit, based on drilling data. There are few structural constraints on this fracture and fault system, as the reservoir is buried beneath more than 100 m of Cretaceous mudstone. At the basin scale, regional faults are considered to have played a role in the transfer of mineralizing fluids during the uranium ore deposition stage (Gerbeaud, 2006; Mamane Mamadou et al., 2022; Pagel et al., 2005).

### 2.2 Lithostratigraphy

The lithostratigraphic column of the Tim Mersoï basin (Figure 1e) is divided in three main sedimentary stages: a fluvio-deltaic period from Devonian to Lower Permian, a continental sedimentation from Permian to Jurassic and a lacustrine sedimentation in the Lower Cretaceous (Valsardieu, 1971). Uranium mineralization in the Imouraren deposit is hosted within the Tchirezrine II sandstone unit, corresponding to the Upper Jurassic part of the Agadez group. This unit shows an average thickness of 50 m and contains arkosic sandstones enriched with reworked analcime (i.e. diagenetic mineral classified as a zeolite and inherited from fine volcanic sediments) intraclasts, as well as massive analcimolite horizon. These sediments represent fluvial deposits in a braided system evolving into a meandering system (Mamane Mamadou, 2016; Valsardieu, 1971). This unit is carrying a confined aquifer. The Tchirezrine II unit overlies the argillaceous (+ analcime and fine-grained sandstones) Abinky formation (Upper Jurassic) and is overlayed by a 100 m thick series of Lower Cretaceous claystones (Assouas siltstone and Irhazer mudstone), above the Imouraren deposit.





**Figure 1: (A)** Location map of the Tim Mersoï basin in North-Niger. **(B)** Geological map of the study area showing the location of the Tchirezrine II outcrop and the Imouraren site. **(C)** Cross-section between Arlit fault and Tchirezrine II outcrop (Orano internal report). **(D)** Location of the studied wells within the Imouraren site, red wells are ones from Figures 7d, 8d and 10c. **(E)** Lithostratigraphy modified from Gerbeaud (2006).


## 3 Material and methods


The approach developed in this study combines well data analysis, including core description, borehole images and geophysical logging, lineament quantification from satellite images, and hydrogeological tests.

### 3.1 Lineament study

In order to better constrain the geometrical organization of the deformation structures affecting the Imouraren reservoir, a

part of this study involves an analysis of lineaments using satellite images.



### 3.1.1 Data acquisition

The coordinate reference system (CRS) EPSG:32632 - WGS 84 / UTM zone 32N was used during this project. Images from Google Earth Pro database have been used to digitalize 5 different sets of fracture networks. Google, Landsat / Copernicus, CNES Airbus and Maxar Technologies assembly of ortho-mosaics was used to create a first-order lineament map to reference

large scale structures affecting the Tim Mersoï basin. This rectangular map (x = 312798; 392132 and y = 1940042; 2041469) covers a sampling area of 7500 km² from the Aïr Massif to the Imouraren site from east to west, respectively. Google, Maxar Technologies ortho-mosaics of 0.3 m/pixel resolution have been used to sample second-order lineaments from 4 different targets in the Tchirezrine II unit. We placed a circular sampling area in each target, following Mauldon et al. (2001) and Watkins et al. (2015) recommendations to inhibit orientation bias. The size of each circle varies and ranges from 72.9 m to 123 m

radius, in order to maximize the sampling area of each target (i.e. each outcrop area has been chosen to optimize the cleanest possible sampling surfaces and to limit the censoring bias). In addition, the size of these circular sample surfaces is of the same order of dimensions as a set of ISR cells.

We have digitized lineaments network using QGIS 3.24 (QGIS Development Team, 2020). Built-in functionalities from QGIS were used to extract topological parameters such as azimuth, length, intersection points (mentioned as *nodes*) and censored

lineaments (e.g., traces which cut the sampling windows or censored areas like covering sand deposit). Azimuth parameter was established for each lineament considering the straight line between the starting and the end point. Length parameter was measured for each lineament from the addition of all segment lengths (i.e. a segment is here defined as a straight line between vertices, and lineaments are made up of a set of segments inter-connected by vertices). Nodes were extracted from the intersection of cross-cutting lineaments.

### 145 3.1.2 Fracture network characterization

*Azimuth sets*

Azimuths from each sampled area were plotted in a length-weighted half rose diagram, where bins represent 10 degrees of azimuth, and the radius is the length-weighted frequency. For the first-order lineaments map (see section 3.1.1), the rose diagram was compared to the large-scale structures described in the literature (Gerbeaud, 2006). This rose diagram is also

useful to have a view of the Tchirezrine II fracture networks within the large-scale structural context of the Tim Mersoï basin. Rose diagrams of second-order lineaments (see section 3.1.1) were used to identify different sets of lineaments at the reservoir scale. These sets can be analysed separately in order to characterize their similarities and differences (i.e. spacing, length distribution, nature).

*General network parameters*

To describe a fracture set distribution, Dershowitz and Herda (1993) have introduced the *fracture intensity* (m$^{-1}$) parameter (*P21*), which is calculated from the total fracture length (*$\Sigma length$*, m) divided by the *sampled area* (m²). In addition to size distribution and spacing, this parameter is an important geometrical attribute to characterize distribution of fracture networks.

*Spacing*

Spacing analysis was done for second-order lineaments sets in the circular sampling area. Each set of orientation was analysed

individually by image processing. One binary image (black and white) by set of lineaments was extracted without the background Earth's surface image and was rotated to position the average azimuth along the *x*-axis of the image. Using the *Analyze Line Graph* tool of ImageJ software (Abràmoff et al., 2004), each pixel column along the *y*-axis (normal to *x*-axis) was analysed to export the coordinates of each lineament intersecting the pixel column. For each column of pixels, we can



therefore calculate the distance between two consecutive lineaments. In order to compare each set to each other, and to
represent the distribution of the spacing values, we have plotted the values in boxplot graphs. The coefficient of variation ($C_v$),
defined as the standard deviation divided by the mean spacing (Cox and Lewis, 1966), was used to discuss the spatial
distribution of each set. According to Odling et al. (1999), if $C_v < 1$, the lineaments are regularly spaced and if $C_v \geq 1$, the
lineaments then show a random to more clustered distribution.

*Length distribution*

Length distribution analysis is commonly used to characterize the geometrical properties of a fracture sets (Cowie et al., 1995;
Jackson and Sanderson, 1992; Soliva and Schultz, 2008; Walsh et al., 1991). This was done by plotting lineament *length* versus
*cumulative frequency* (Childs et al., 1990) and quantifying the fit law of the distribution trend, with its least square coefficient
($R^2$). A power-law distribution, which is commonly found in fault length distribution, was described as scale invariance in
lineaments length (Watterson et al., 1996; Yielding et al., 1996), whereas exponential distribution characterizes scale
dependence. Scale dependence is generally described for joint set see Bai et al. (2000) due to mechanical layering (i.e. lithology
or structural boundaries) that limits the structure propagation (e.g., for lithological fault ending see Soliva et al. (2006)).

To perform a length distribution analysis, two geometrical biases are important to be determined. (i) Truncation bias affects
the frequency of small lineaments due to the limited resolution of the orthophotography in which the lineaments are detected
(i.e. the smallest fractures will be under sampled due to the image resolution). This bias can be considered by using a lineament
size cut-off below which lineaments are too much truncated and should not be included in the determination of the size
distribution law (Heffer and Bevan, 1990; Odling, 1997). In this paper, in accordance with other studies (e.g., Bonnet et al.,
2001; Soliva and Schultz, 2008) and the resolution of satellite images, we have used a truncation cut-off of 6 meters in
lineament length. (ii) Censoring bias is the underestimation of the length of generally large lineaments caused by the limitation
of the sampling window or by sand cover, in our specific case. Following recommendation from Yielding et al. (1996), we
have included lineaments that are affected by censoring in the determination of the size distribution law since excluding them
gives more error in the scaling determination.

**3.2 Wells data**

The data set used for this study was carried out from 12 vertical boreholes, including core descriptions and geophysical logging.
The study focuses on boreholes drilled in the southern part of the Imouraren deposit, spaced from a hundred meters to several
kilometres apart. The data set presented in this study comes mainly from the basal section of the reservoir, between 133 m and
160 m depth, consistently with the scope of the Uranium ISR project.

**3.2.1 Core description**

Drill cores have been described in term of deformation features, lithology and granulometry, i.e. from very fine sandstone
(VFSs) to very coarse sandstone (VCSs) using conventional grain diameter classes (Nichols, 2009). As this study focused on
sandstones, data classified as VFSs, showing systematically high proportion of clay, were excluded from the structural analysis.

**3.2.2 Optical borehole images**

Deformation structures have been classified from analyses of optical boreholes images (OBI, based on colour and aspect) and
double checked from drill-core pictures based on apparent morphology (e.g., texture, shear displacement, aperture and
cataclasis). From these observations, structures have been classified in three types: Mode I fractures, deformation bands and



faults following the terminology of Fossen, (2016). *P10* density is calculated following Dershowitz, (1984), corresponding to the number of fractures counted per meter (m$^{-1}$) along a 1D scanline, here corresponding to the borehole.

Geometrical attributes of these structures, such as dip and azimuth can be extracted from OBI, by fitting a sinusoidal curve on unrolled and oriented images (Zemanek et al., 1970). Orientation data from picked structures have been analysed only with the azimuthal component in order to compare it with lineament data from 2D satellite images.

### 3.2.3 Geophysical logging and processing

Logging data consist in:

(i) Waves slowness of the formation as well as water slowness from Full-Wave-Sonic logging-tool with porosity estimation (*PHIS*) based on Wyllie et al. (1956) equation (1):

$$PHIS = \frac{Dtc - Dtc_m}{Dtc_f - Dtc_m} \tag{1}$$

Where *Dtc* is formation slowness (µs/m), *Dtc_m*, matrix slowness (µs/m) set at 173 µs/m for the study and *Dtc_f*, the fluid slowness (µs/m).

(ii) A permeability log from Nuclear Magnetic Resonance (NMR) logging-tool with permeability estimation (*K_SDR*) based on Schlumberger-Doll-Research with equation (2) (see Elsayed et al. (2022) and Hidajat et al. (2004) for more information on NMR theorical background):

$$K_{SDR} = a \times (T_{2LM})^2 \times TPOR^4 \tag{2}$$

Where *K_SDR* is the NMR permeability, *a*, a formation-dependent variable, *T_2LM*, the logarithmic mean of the *T_2* relaxation time (ms), and *TPOR*, total porosity from NMR measurement (%).

A sampling step of 0.1 m was used to recover both *PHIS* and *K_SDR* across the reservoir. Each data is associated with lithological label defined from core description and plotted in porosity-permeability graph.

We used the derived Kozeny-Carman equation (3) from Bear (1972) to model the theoretical evolution of permeability as a function of porosity for various homogenous grain size diameters. Different theorical porosity-permeability curves corresponding to various grain size classes are used for comparison with measured dataset.

$$k = \frac{(\varphi^3 \times d^2)}{180(1-\varphi)^2} \times (1 \times 10^{12}) \tag{3}$$

Where *k* is the permeability (Darcy), *φ*, the porosity (%) and *d*, the grain size diameter (m).

### 3.3 Aquifer testing

An aquifer testing has been realized on well IMOU_2527-2 in order to estimate the characteristics of a depression cone in long-duration pumping operation. Two piezometers were drilled at 17 m from IMOU-2527-2 in the south (IMOU_2527-3) and east (IMOU_2527-4) directions. During the pumping sequence, the piezometric level was measured manually every minute for the first 10 minutes and with an increasing sampling step to reach a measurement every 2 hours after 72 hours. The test was stopped after 830 hours (~35 days) of pumping. The final dewatering levels in the piezometers have been used to estimate the extension of the cone of depression in the east and south directions by plotting these levels (m) versus the distance of the piezometers (m) in logarithmic scale. A logarithmic trendline fit has been made to show the distance at which dewatering reaches 0 m.



Tracing involves injecting a tracer (NaCl brine) into a piezometer located in the drawdown cone of a pumped well into production and observing its recovery at the pumping well. Forced flow tracing was used, involving the injection of 1700 l of fresh water after the brine injection. Monitoring was both carried out on the surface, by installing an on-line conductivity meter on the discharge and by in situ resistivity log (the recording interval was 1 hour for the duration of the tracing). Salt restitution, and in particular chloride restitution, is assessed by establishing a correlation line between [Cl-] vs. conductivity. The chloride

restitution is then estimated over the time. (additional information on tracing operations are exposed in a table in supplementary material 1). Tracing was firstly performed in N-S axis (from IMOU_2527_3 to IMOU_2527_2) and then in E-W axis (from IMOU_2527_4 to IMOU_2527_2) after returning to the initial conductivity condition.

## 4 Results

### 4.1 Lineaments analysis from satellite images

#### 4.1.1 Basin-scale structures

At the basin scale, the lineaments are organized in four main sets of distinct orientations (Figures 2a and b):

(i) The N060°E-N080°E (ENE-WSW) set consists of lineaments with apparent dextral shear movement (Figures 2d and e). These faults are distributed homogeneously across the sampling window although they are sometimes difficult to observe because of the quaternary sand cover (Figures 2a and c). Some minor E-W trending faults are commonly observed linked to

250 these ENE-WSW faults (Figures 2c and e);

(ii) The N110°E-130°E (ESE-WNW) set is composed of lineament with apparent sinistral shear movement (Figure 2d). These faults are conjugate to the ENE-WSW dextral strike-slip set as illustrated by their mutual crosscutting relationships (Figures 2c and d);

(iii) The N170°E-N010°E (N-S) set is sub-parallel to the Arlit fault that borders the western part of the Imouraren deposit

(Figure 2f). This set is poorly outcropping because of the sedimentary cover (Irhazer claystone as well as actual aeolian and alluvial deposits) and frequently underlined by the presence of folds on the western part of the study area;

(iv) The set N010°E-N040°E (NNE-SSW) is mainly detected in the basement of the Aïr Massif. This set also affects the sedimentary cover but is generally expressed by km-scale folds related to the faults. Two main structures named Madaouela (Figures 2a - north one and d, and Figure 1b) and Adrar Emoles (Figure 2a - south one) are present in the study area. These

260 structures have about 40 km of spacing and crosscut both the basin and the basement from the eastern part to the western part of the window. The Madaouela structure is also bordering the northern part of the Imouraren deposit and links to the Arlit N-S fault (Figure 1b).





**Figure 2: Lineaments map and their features observed at the basin scale. A) Mapping of lineaments at the basin scale. B) Rose diagram showing the length-weighted azimuth of mapped lineaments. C) Detailed view of the large-scale map focusing around the Imouraren site and the outcropping Tchirezrine II area. The location of the Z1 to Z4 sampled areas is also reported on this map. D) Example of the conjugate system of N070°E and N110°E strike-slip faults cutting across a N030°E trending fold. E) Example of E-W faults associated to a N060°E fault. F) Example of N010°E trending fold related to a segment of the Arlit cluster faults. Maps data: © Google (A, C, D, E and F),** *Landsat / Copernicus (***A), Airbus (C and D)***, CNES* **/ Airbus (C), Maxar Technologies (C, E and F).**

### 4.1.2 Detailed lineament networks in the Tchirezrine II

A total of 4779 lineaments were mapped from the four circular sampling windows selected in the Tchirezrine II (Figure 3 and Table 1). These data were used to characterize the structural organization of lineament networks affecting the Tchirezrine II reservoir at the scale of ISR project, i.e. ranging from meter to hundred meters scales.





**Figure 3: Detailed description of the lineaments affecting the Tchirezrine II in the four circular sampling windows (Z1 to Z4). (A) Satellite images showing the sampling window in red. (B) Lineament traces inside the window. (C) Length-weighted rose plot showing lineament azimuth distribution. The red lineaments are censored at the boundary of the sampling window. See Figure 2c for the location of sampling windows in the study area. Satellite images data from © Google, Airbus (Z1, Z2, and Z3) and Maxar Technologies (Z4).**



| SPACING | NE-SW | | | | | NW-SE | | | | | E-W | | | | |
|---|---|---|---|---|---|---|---|---|---|---|---|---|---|---|---|
| | Z1 | Z2 | Z3 | Z4 | Av | Z1 | Z2 | Z3 | Z4 | Av | Z1 | Z2 | Z3 | Z4 | Av |
| **Mean** | 7.5 | 7.0 | 6.5 | 5.8 | 6.7 | 7.6 | 5.1 | 5.1 | 4.8 | 5.7 | 11.3 | 19.0 | 18.9 | 8.9 | 14.5 |
| **Median** | 6.5 | 5.8 | 5.1 | 4.6 | 5.5 | 6.8 | 3.9 | 3.9 | 3.7 | 4.6 | 8.4 | 11.3 | 11.4 | 6.5 | 9.4 |
| **Sd** | 4.4 | 4.5 | 4.6 | 3.9 | 4.4 | 4.4 | 3.7 | 3.8 | 3.5 | 3.9 | 9.1 | 20.5 | 22.1 | 7.7 | 14.9 |
| ***Cv*** | 0.6 | 0.6 | 0.7 | 0.7 | 0.65 | 0.6 | 0.7 | 0.7 | 0.7 | 0.68 | 0.8 | 1.1 | 1.2 | 0.9 | 1.02 |
| **nbr** | 2129 | 6386 | 7817 | 7320 | | 3931 | 9780 | 12146 | 8848 | | 3037 | 2083 | 3079 | 4947 | |
| **LENGTH** | | | | | Zall | | | | | Zall | | | | | Zall |
| ***Σlength*** | 1055 | 3725 | 6484 | 2727 | 13991 | 1626 | 5277 | 9217 | 3321 | 19441 | 1245 | 1266 | 2479 | 2761 | 7750 |
| **Mean** | 11.5 | 7.7 | 11 | 13.2 | 10.2 | 18.1 | 9 | 12.6 | 11.8 | 11.5 | 20.1 | 9.4 | 14.2 | 16.4 | 14.4 |
| **Median** | 9.3 | 5.8 | 9.5 | 11.1 | 8.5 | 14.1 | 6.5 | 11 | 8.9 | 8.9 | 12.1 | 8 | 12.1 | 13.9 | 11.5 |
| **Sd** | 7.5 | 6.1 | 5.9 | 8.7 | 6.9 | 14.2 | 8.9 | 7.1 | 9.5 | 8.9 | 19.8 | 6 | 8.5 | 10.6 | 11.1 |
| **nbr** | 92 | 485 | 588 | 207 | 1372 | 90 | 585 | 731 | 282 | 1688 | 62 | 135 | 174 | 168 | 539 |
| **EXP** e | -0.13 | -0.15 | -0.17 | -0.11 | -0.15 | -0.06 | -0.09 | -0.14 | -0.10 | -0.11 | -0.04 | -0.16 | -0.11 | -0.09 | -0.08 |
| **R²** | **0.99** | **0.99** | **0.99** | **0.99** | **0.99** | **0.99** | 0.92 | **0.98** | **0.99** | **0.99** | 0.93 | **0.99** | **0.99** | 0.98 | **0.99** |
| **PL** e | -1.90 | -2.25 | -2.35 | -1.86 | -2.20 | -1.42 | -1.79 | -2.05 | -1.74 | -1.88 | -1.30 | -2.22 | -2.01 | -1.75 | -1.81 |
| **R²** | 0.93 | 0.95 | 0.91 | 0.87 | 0.92 | 0.91 | **0.99** | 0.87 | 0.92 | 0.93 | **0.97** | 0.96 | 0.85 | 0.78 | 0.89 |

**Table 1: Lineament spacing (m) and Length (m) attributes from statistical analysis. The average value (Av.) is calculated from the mean value of each sampling window. *Σlength* is equal to the cumulative length of set in the sampled window. "Zall" represents merged length data sets. Exponential (EXP) and Power-Law (PL) distribution fits are specified with their exponent (e) and Coefficient of Determination (R²).**

*Azimuth sets*

For all the studied circles, two main sets of lineaments trending N030°-060°E (NE-SW) and N110°-140°E (SE-NW) are detected. These two almost orthogonal sets represent 24.8 % and 35.4% of the total length-weighted trace azimuths, respectively. A third set trending N070°-100°E (E-W) is detected with a proportion of 13.7 %. However, this proportion is

290 variable from a sampling area to another, i.e. high proportions of 21.8 % and 26.3 % in the Z1 and Z4 areas, and low proportions of 8.9 % and 9.5 % in the Z2 and Z3 areas respectively. A minor set trending N160°-010°E (N-S) is also detected in the Z3 area, with a proportion of 13.5 % (this set is not described in detailed in the following result section).

*Spacing*

All the three sets of lineament trends described above show high spacing values due to the extensive sand cover in the study area (see the sandy surfaces visible in Figures 3 Z2 and Z3). To improve the representativeness of the data, we manually checked the largest spacing values in places without censoring bias for each set of lineaments and ignored higher spacing values due to censoring by sand cover. For the two main sets (i.e. NE-SW and NW-SE), we found that the actual value of the largest spacing is close to 20 meters in Z1, while for the E-W set, we were unable to find a specific value (i.e. which is lower than censored ones).

than censored ones).

The average median spacing of the NE-SW set is 5.5 m (Table 1) and ranges from 6.5 m for Z1 to 4.6 m for Z4 (Figure 4a). The NW-SE set average median spacing is 4.6 m (Table 1) and ranges from 6.8 m for Z1 to 3.7 m for Z4 (Figure 4b). Then, the NE-SW set shows higher median spacing values compare to NW-SE, for all the four-sampling window, with an average factor of 1.2. Spatial distributions for the two main sets are characterized by the coefficient of variation ($C_v$) that shows values

between 0.6 and 0.7, which are correlated with regularly spaced lineaments (Odling et al., 1999). Compared to the two main sets, the spacing distribution of the E-W trending set shows larger and more heterogeneous spacing values (Figure 4c). Data from locations Z2 and Z3 show a median spacing ranging from 11.3 m and 11.4 m, with a standard deviation ranging from 20.5 to 22.1 whereas data from locations Z1 and Z4 show median values of 8.4 m and 6.5 m and a relatively low standard deviation of 9.1 and 7.7 (Table 1). This set shows an average coefficient of variation of 1.02 that is correlated with random to

clustered lineaments pattern. For this orientation set, Z1 and Z4 have $C_v$<1, which indicates more regularly spaced pattern of lineaments.



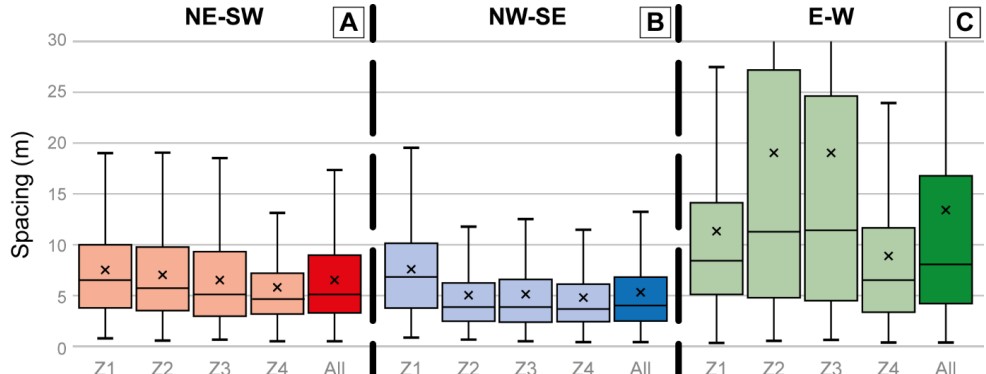

**Figure 4: Boxplots showing lineament spacing data from the NE-SW (A), NE-SW (B) and E-W (C) sets. Boxplots mentioned as "*All*"**
**are representing merged datasets from all zones (Z1 to Z4). Bottom of a box is the 1ˢᵗ quartile and the top is the 3ʳᵈ quartile, horizontal**
**line inside the box is the median value and the cross is the mean value. Vertical lines represent the interval of 95% of data.**

*Length distributions*

Considering the merged data from the 4 sampling windows, the best fit of length distribution for the three different trending
sets is an exponential law, with high determination coefficients, i.e. $R^2$ of 0.99 (Table 1 and Figure 5). Some data from all sets
are subject to censoring, which is especially true for the E-W set. For example, the fault segment from Z1, sampled with a
length of 133 m, shows a minimum lateral continuity of 1080 m (Figure 3a). Some of the longer E-W lineaments are censored,
with over 28% of the *Σlength* of the E-W merged set subject to censoring, more than the NE-SW and NW-SE sets, which are
at 23% and 22%, respectively.

The exponential exponent of the NE-SW merged set is -0.15, while that of the NW-SE set is -0.11 (Table 1). This difference
in distribution shows that maximum length values are larger for the NW-SE merged set. The exponential exponent of the E-W
lineaments is -0.08, describing a much larger maximum length compared to the two main sets, with lineament length values
closer to the window sizes (Table 1 and Figure 5c). Note the linear shape of the data alignment at larger scale on the log-log
graph (Figure 5c) and that these data are more censored than the two other sets (Figure 5b and c), revealing more scale
invariance of this E-W set than the two others.

Considering the different zones separately allows to highlight the impact of large-scale structures on the lineament length
distribution. As shown above, the E-W set for Z1 location (Figure 3-Z1) shows a length distribution that is better following
scale invariant trend than the other sampled zones (Table 1). Similarly, the Z1 and Z2 locations are bordered by faults (Figure
3a, Z1 and Z2), shows a length distribution of the NW-SE set that follows a scale invariant trend with large exponential
exponent (Table 1, -0,06 and -0,09).



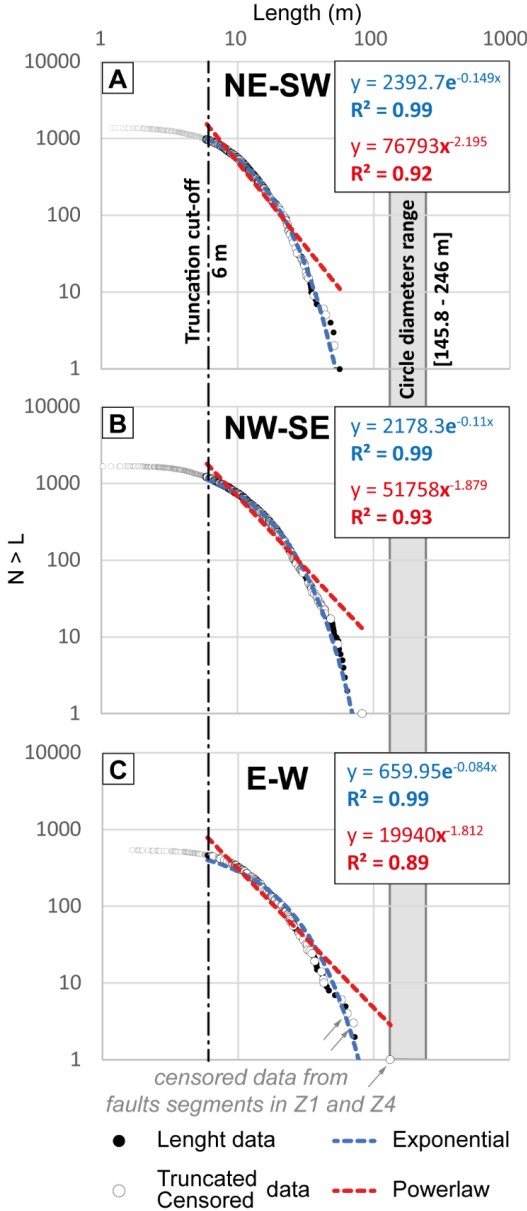

**Figure 5: Graphs showing the length distributions of lineaments from the NE-SW (A), NE-SW (B) and E-W (C) sets. These data are merged from Z1 to Z4 sampled areas. Truncation cut-off is represented by the black dashed-line whereas the Exponential and Power Law best fits are drawn by a blue and a red dashed-line, respectively.**


*Fracture intensity and Connectivity*

Considering the total dataset, the average fracture intensity *P21* is 0.48 m$^{-1}$ and the apparent connectivity is 0.09 nodes/m². The maximum *P21* value equals 0.59 m$^{-1}$ in the Z4 and the minimum *P21* value equals 0.34 m$^{-1}$ in the Z1. The same repartition is observed for apparent connectivity with a maximum of 0.14 nodes/m² in the Z4 and a minimum of 0.04 nodes/m² in the Z1.

Considering these parameters by trending sets separately, the NW-SE *P21* is always higher than NE-SW *P21* (0.04 to 0.05 m$^{-1}$) for all zones. The average E-W *P21* of 0.07 m$^{-1}$ is lower than the other sets but is noticeably high in Z4 with 0.16 m$^{-1}$.





Connectivity is proportionally increasing with *P21* of NE-SW and NW-SE sets, with an apparent linear trend for Z1, Z2 and Z3 (Figures 6a and b). This relationship suggests that lineament connectivity is dominated by these trending sets in these 3 sampling windows. Data from Z4 (table 2) show a high connectivity of 0.14 that is not related to an increase of *P21* of NE-

SW and NW-SE lineaments. In this zone, the high E-W *P21* value (0.16 m$^{-1}$) significantly increases the fracture connectivity.

|  | Z1 | Z2 | Z3 | Z4 | Zall |
|---|---|---|---|---|---|
| *Connectivity* (nodes/m²) | 0.04 | 0.08 | 0.10 | 0.14 | 0.09 |
| *P21* (m/m²) | 0.34 | 0.41 | 0.55 | 0.59 | 0.48 |
| NE-SW *P21* | 0.06 | 0.11 | 0.14 | 0.15 | 0.12 |
| NW-SE *P21* | 0.10 | 0.15 | 0.19 | 0.19 | 0.17 |
| E-W *P21* | 0.07 | 0.04 | 0.05 | 0.16 | 0.07 |

**Table 2:** *Connectivity* **and** *fractures intensity (P21)* **for all sampling window and merged data set (Zall).** *Connectivity* **is calculated from nodes number and window radius mentioned in Figure 3.** *Fractures intensity* **is calculated from** *Σlength* **(Table 1) and window radius.**

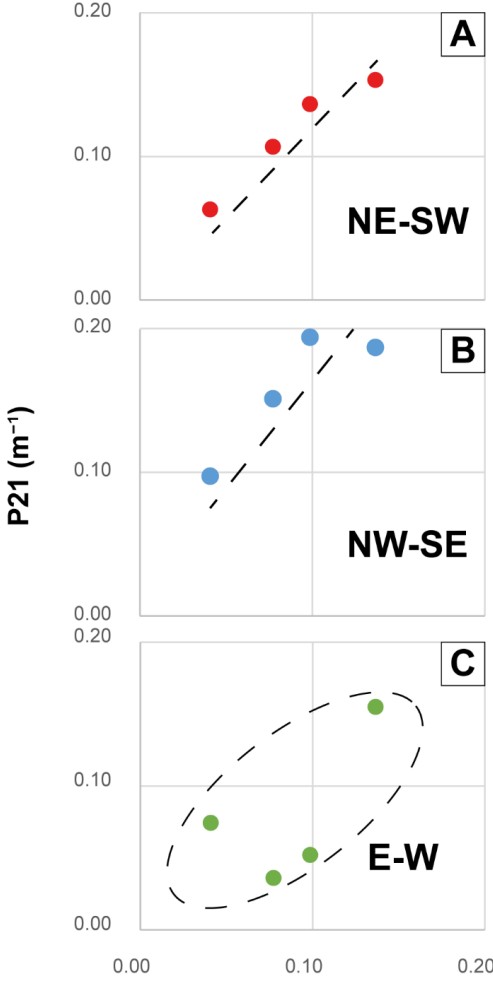

**Figure 6. Graphs of** *Connectivity* **(nodes/m²) versus** *fractures intensity* **(*P21*) (m$^{-1}$) from the three trending sets for Z1 to Z4. General**
**trends are highlighted by black dashed lines. A) NE-SW sets, B) NW-SW sets and C) E-W sets.**



### 4.2 Structures description from wells

Deformation structures have been classified into three main types, using both OBI data and core description (see Figure 7 and supplementary material 2 for additional illustration of structures): (i) Mode I fractures, (ii) Deformation bands, and (iii) Faults.

(i) Mode I fractures are the most observed structures with a total of 256 fractures picked on OBI data. They are mainly trending
E-W with 66 % of them in an azimuth range between N080°E and N110°E and dip sub-vertically to the North and South, at an average angle of 78.9° ± 10.3°. These fractures show sharp cut edges and are open or sealed. Dark edgings are observed for open fractures whereas cements, generally clays or oxidized products, yellow uranium products, harmotome (Barium silicate) or carbonates precipitation, are detected in sealed fractures (Figure 7a, and supplementary material 2 for additional examples)

(ii) 162 deformation bands are observed on OBI as white linear structures, a few millimetres to a centimetre thick (Figure 7b,
and supplementary material 2 for additional examples). Crushed grains and cataclastic textures are observed inside these bands. They are composed of a single or several anastomosed strands. These bands are sometimes crosscut or open by Mode I fractures. They can be classified as cataclastic compactional-shear bands. They are mainly trending E-W with 71 % of them in an azimuth range between N080°E and N110°E and dip sub-vertically to the north and south, at an average angle of 77.7° ± 10.9°.

(iii) Faults are observed as zones of cataclastic rock and crush breccia (Figure 7c). These fault rocks are surrounded by zones of high density of Mode I fractures and cataclastic deformation bands representing the inner-fault damage zone (DZ). These faults are observed in 2 wells located in the southern part of the Imouraren site (Figure 1d). A single fault core dipping to the south is observed in IMOU_1471_2 well. The IMOU_0382_2 well shows 3 distinct fault cores, dipping to the north, over a more than 20 m thick fault zone. These faults picked on OBI show an azimuth ranging between N050°E and N080°E (Figure
7c). Almost all the Mode I factures and cataclastic deformation bands picked on OBI within these fault DZs are trending E-W. Scarce field observation are possible, they are consistent with the structures observed in OBI, although some Mode I fractures, and deformation bands are also observed trending N070°E on outcrop (supplementary material 2).

Both fractures and deformation bands are observed in nearly all sandstone facies, i.e. there is no significant correlation between facies and observed structure type (Figure 7b). Scarcely, a single structure can be observed across different facies. These rare
cases reveal changes in morphological characteristics, e.g. deformation bands tend to be finer in fine-grained sandstones and thicker in coarser sandstones but remain cataclastic deformation bands.

The *P10* density of structures appears to be very heterogeneous from one borehole to the next (Figure 7d), ranging from 0 to 4.55 m⁻¹ (Table 3). Positions of the structures along boreholes are reported using the mean of OBI sinusoid. The highest densities of structures are observed in boreholes intersecting fault cores (Figure 7c) or close to N070°E master faults (Figure
1d), such as IMOU_0236_4 or IMOU_0182_3 (Table 3 and Figure 7d). On the other hand, note that the borehole IMTS_0115_2 reported in Table 3, does not intersect any structures.



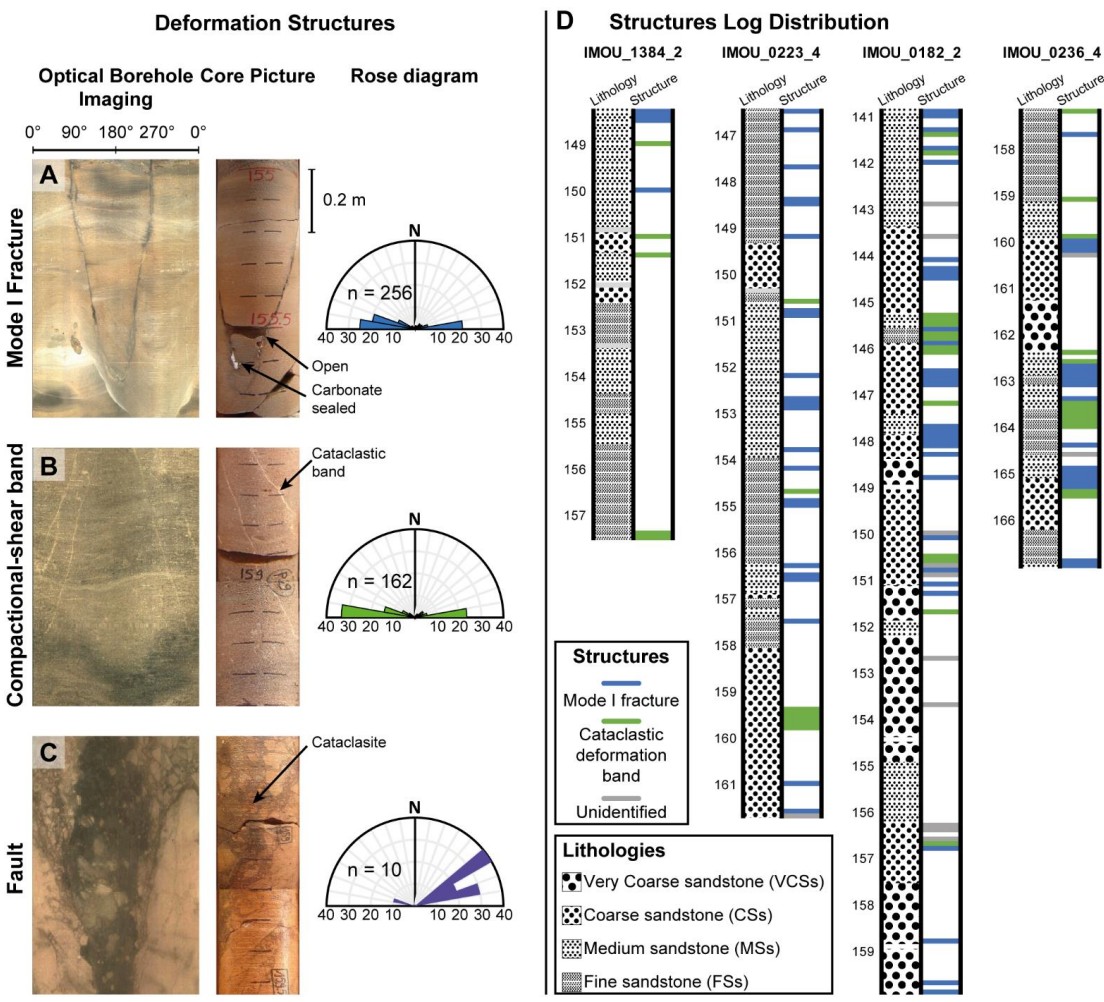

**Figure 7: Deformation structures detected from well data in the Tchirezrine II reservoir shown with core sample image, OBI data and corresponding rose diagram for each type of structure. (A) Open or sealed Mode I fractures. (B) Cataclastic compactional-shear bands. (C) Fault core containing cataclasite or crushed breccia. (D) Detected structures from four wells with low to high degree of fracturing. See associated *P10* density and Porosity/Permeability distribution of these 4 wells in Table 3 and Figure 8d, respectively.**

| Well ID | P10 (m⁻¹) |
|---------|-----------|
| IMOU_0182_3 | 2.94 |
| IMOU_0223_4 | 2.01 |
| IMOU_0236_4 | 3.39 |
| IMOU_0382_3* | 3.18 |
| IMOU_0400_3 | 2.07 |
| IMOU_1228_2 | 2.78 |
| IMOU_1278_2 | 0.62 |
| IMOU_1384_2 | 0.99 |
| IMOU_1471_2* | 4.55 |
| IMOU_2527_2 | 1.68 |
| IMTS_0115_2 | 0.00 |

**Table 3: All type of structures *P10* density within wells with optical borehole image, see Figure 1d for the location of the wells. (*) Boreholes intercepting fault cores.**

### 4.3 Porosity and Permeability Relationships

In this section, we present the porosity and permeability values obtained by the well log measurements from the in-situ Imouraren reservoir (Figure 8a). These values are firstly described considering the whole sonic porosity and NMR permeability dataset. These data are secondly described considering grain size of the corresponding sandstone layer defined during the core description, i.e. from fine sandstone (FSs) to very coarse sandstone (VCSs) (Figure 8a). Reference curves calculated using Kozeny-Carman equation (eq. 3) are plotted on the graphs in order to compare these trends with the logging data sorted by granulometry (Figures 8b and c). Thirdly, these data are described using 4 different wells selected as a function of the *P10* density of fractures described on OBI picking (Figure 8d).



Considering the total data set, the sonic porosity of this reservoir is ranging from 4% to 31% (Figure 8a and Table 4). The average value is of 19 % with a standard deviation of 4.8 % (Table 4). The NMR permeability of the Imouraren reservoir ranges over five orders of magnitude, i.e. from 0.01 mD to 3 D (1E-10 to 3E-05 m.s$^{-1}$) (Figure 8a and Table 4). The average value is 135 mD (1.35E-06 m.s$^{-1}$) with a standard deviation of 240 mD (2.4E-06 m.s$^{-1}$) (Table 4). This dataset shows a poorly defined normal correlation between porosity and permeability values. Lower permeability values (< 1 mD or 1E-08 m.s$^{-1}$) are

mainly obtained in low porosity materials (<15%). In contrast, the largest permeability values (> 1 D or 1E-05 m.s$^{-1}$) are reached in material of only moderate porosity values (15-20%). The highest porosity values (>25%) are correlated with moderate permeability ones (5 to 500 mD or 5E-08 to 5E-06). The Kozeny-Carman relationship, describing the porosity-permeability relationships of homogeneous granular materials follows a power-law-like trend (Kozeny, 1927; Carman, 1937, 1956). For such a law the data set shows a least squares coefficient (R²) of only 4·10-5 (Figure 8a).

A main observation of this dataset is that there is no relationship between lithology (granulometry) and porosity-permeability (Figure 8a). There is a slightly higher average porosity as a function of granulometry increase, from 17.6% in fine-grained sandstones (FSs) to 20 % in very coarse-grained sandstones (VCSs) (Figure 8a), but the range of porosity values is almost similar from one lithology to the other (i.e. from 6.5 % to 26.1 % for FSs and from 6 % to 29.5 % for VCSs, Table 4). Most of the data do not follow the expected trend of porosity-permeability Kozeny-Carman trend relationship, expressed by reference

curves for each grain size class (Figures 8b and c). For the FSs, only 13.3 % of porosity data ranges in the corresponding Kozeny-Carman area (green area in Figure 8b) whereas 77 % of these data show higher permeability for a given porosity (Figure 8b). These permeability values can surpass the theoretical curves by one or two orders of magnitude. For the VCSs, only 8.1 % of the data range in the corresponding Kozeny-Carman area (red in Figure 8c) whereas 91.6% of these data show lower permeability for a given porosity (Figure 8c). These permeability values can be up to three orders of magnitude lower

than the theoretical values expected for a very coarse-grained sandstone.

The second main observation about this dataset is that its permeability distribution is highly influenced by fracture density. Fracture *P10 density* is positively correlated to the permeability values, whereas no clear correlation is detected with porosity (Figure 8d). The highest permeability values are obtained for the wells showing a high fracture density (Figure 8d, IMOU_0236_4, and a part of the IMOU_0182_3), whereas the lowest permeability values are obtained for the wells showing

a low fracture density (Figure 8d, IMOU_1384_2, and a part of the IMOU_0223_4). The data from the well with the highest fracture density correspond to the area of Kozeny-Carman trends calculated for coarse grained material (Figure 8d - red cluster). On the other hand, the data from the well with the lowest fracture density correspond to the area of Kozeny-Carman trends calculated for fine-grained material (blue cluster). Consistently, data from wells with intermediate fractures density correspond to intermediate reference trends (yellow cluster).

| | Sonic porosity (%) | | | | NMR KSDR permeability (Darcy) | | | | |
|---|---|---|---|---|---|---|---|---|---|
| | *Mean* | *Sd* | *Min* | *Max* | *Mean* | *Sd* | *Min* | *Max* | *n* |
| **FSs** | 17.6% | 4.0% | 6.5% | 26.1% | 1.35E-01 | 2.07E-01 | 2.89E-05 | 1.52E+00 | 592 |
| **MSs** | 18.8% | 5.8% | 4.2% | 31.4% | 1.11E-01 | 2.36E-01 | 5.41E-07 | 2.84E+00 | 598 |
| **CSs** | 19.5% | 4.4% | 4.9% | 30.4% | 1.69E-01 | 3.21E-01 | 8.26E-05 | 3.23E+00 | 658 |
| **VCSs** | 20.0% | 4.2% | 6.0% | 29.5% | 1.21E-01 | 1.57E-01 | 4.13E-04 | 8.00E-01 | 315 |
| **AllSs** | 19.0% | 4.8% | 4.2% | 31.4% | 1.35E-01 | 2.44E-01 | 5.41E-07 | 3.23E+00 | 2163 |

**Table 4: Summary of Sonic porosity and *NMR permeability* values considering the total dataset. *Sd* for Standard Deviation. The vertical sampling step is 0.1 m with *n* the number of samples. The cumulative thickness of sampled reservoir is 216.3 m from 12 wells. 1 Darcy = 1E-05 m.s$^{-1}$.**



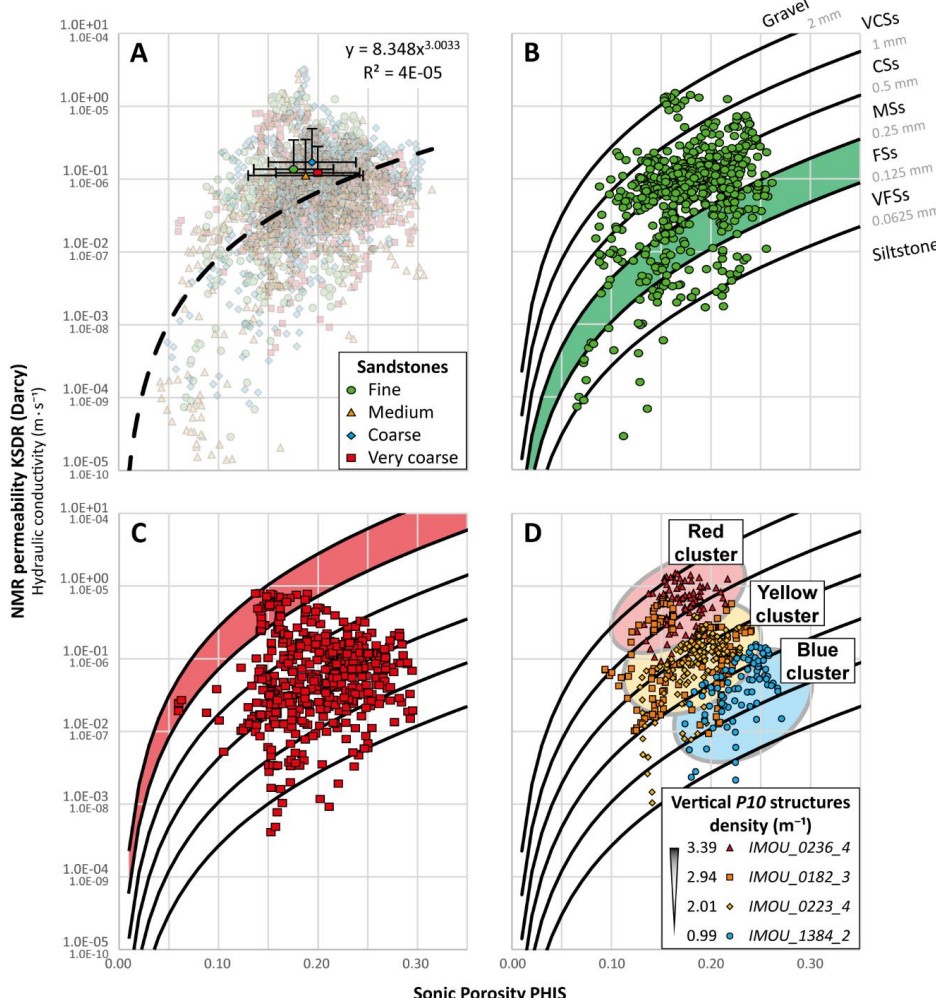

**Figure 8: Graphs of sonic porosity (*PHIS*) versus NMR permeability (*K_SDR*) in the Imouraren reservoir. (A) Graphs showing the whole dataset from the 12 wells, data are sorted by grain size classes obtained by core description, black dash-line represents the power-law trend fit of the whole dataset. The four highlighted points are mean values for each grain size class, with standard deviation as error bars. (B), (C) Graphs showing data from fine-grained sandstones and very coarse-grained sandstones respectively. Reference Kozeny-Carman relationships are plotted for various grain diameters (in mm), the grain size class corresponding to the plotted dataset (0.125 – 0.25 mm for FSs and 1 – 2 mm for VCSs) is coloured in green and red, respectively. (D) Graph showing data from four wells selected for their different vertical fracture occurrences (expressed by their average fracture *P10* density in the legend of the graph, see Figure 1d for the location of the wells, Figure 7d for the structures distribution along borehole of these dataset and Table 3 for all *P10* data). Coloured areas are clusters of data showing differences between high and low fracture occurrences. A vertical sampling step of 0.1 m is used for the whole dataset.**

## 4.4 Aquifer testing

During the pumping phase, the response of the groundwater table is different from the eastern and southern piezometers, from 3 to 12.1 minutes, respectively. At the end of the pumping test (t = 830 h), the aquifer table reaches a lowered level in the pumping well of 72 m. The dewatering in eastern piezometer reaches 41.5 m whereas only 33 m of dewatering is obtained in the southern one. Using a logarithmic fit, we estimate an influence of pumping of 880 m in the south direction and 8670 m in the east direction (Figure 9a). This is approximatively a 1/10 ratio in radius extension. This test revealed a significant anisotropy of the reservoir response following the direction, with a higher drainage pattern along the E-W direction compared to the N-S one.



The restitution curves for the two tracings (Figure 9b) are very different. Tracing from south to north shows a very slow and

diluted restitution, with the first arrivals observed after 87 hours and a chloride deviation that does not exceed 11 mg/l. Inversely, tracing from east to west shows a very rapid and concentrated restitution, with the first arrivals observed after 9 hours and the peak of chloride deviation reaching almost 150 mg/l and occurring after 19 hours.

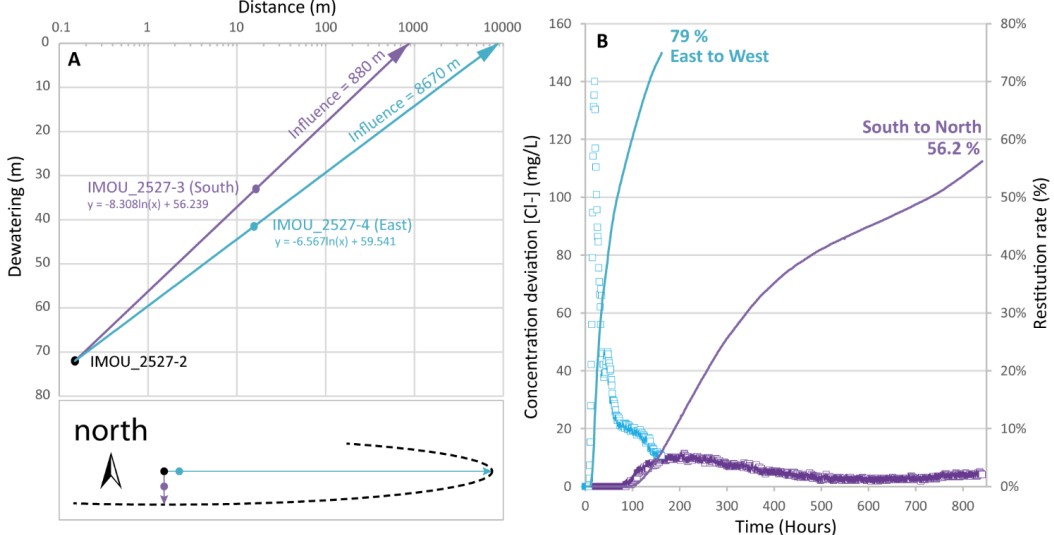

**Figure 9: (A) Graph with drawdown data at *t* = 830 h versus the distance of the piezometers in log scale. Two logarithmic regression**
**curves pass through the point of the well IMOU_2527-2 and go respectively towards the piezometer IMOU_2527-3 in purple and the IMOU_2527-4 in blue. Plan view of the estimated extension of the drawdown cone. (B) Chloride restitution curves over time for the two tracings. (purple) IMOU_2527_3 to IMOU_2527_2 and (blue) IMOU_2527_4 to IMOU_2527_2. Lines for the cumulative restitution rate and squares for the deviation from the initial chloride concentration of the pumped water.**

## 5 Discussion

### 5.1 Main results and limitations

A multifaceted understanding of characteristics and behaviour of the Tchirezrine II reservoir in Imouraren deposit has been obtained through this multidisciplinary approach, and can be summarized in 4 key points:

(i) Detailed analysis from satellite images revealed two main sets of lineaments (NE-SW and NW-SE) with sub-orthogonal trends and regularly spaced patterns (Figure 4), and a third E-W set with lower *P21* density and distribution of length and
spacing characteristics revealing a rather heterogeneous pattern. The lineament connectivity increases significantly where the E-W set is well expressed (Figure 6). It is worth mentioning that length distribution of the NE-SW and NW-SE trend sets are real properties of the lineament network since they are not related to censoring bias of the sampling windows (Figures 5a and b, see supplementary material 3). However, censoring bias seems much more effective on the E-W trend (Figure 5c), having a larger exponential exponent (Table 1) and therefore more scale independent behaviour. Also note that on all the presented data,
truncation bias (section 3.1.2) has little effect on the scaling properties since it is much lower than the main bend observed on the scaling law (Figures 5a and b). It might be more significant on the E-W lineament trend in which the main trend of the curve is closer to the truncation limit (Figure 5c). This is also consistent with a more scale-independent behaviour of this lineament set (see section 5.2 for interpretation). In addition, the presence of undetected fractures, below the resolution of satellite images, may lead to a slight overestimation of the average spacing obtained in the present study, without significantly
affecting their distribution.



(ii) The Tchirezrine II reservoir is affected by Mode I fractures (still open or sealed), cataclastic deformation bands and brecciated-cataclastic faults. The main structural trend observed in OBI, whatever their typology, is oriented E-W (Figures 7a and b). This is fully consistent with the scale independency mentioned before for the E-W lineament trend. Also note that these E-W structures are also observed in OBI in the DZs of N070°E faults (Figures 1d and 7c). Few NE-SW and NW-SE Mode I fractures are observed in OBI, therefore in much lower proportion compared to E-W structures. The vertical configuration of the wells implies a strong sampling bias of the fracture patterns that are almost sub-vertical (Terzaghi, 1965) and have large spacing values (i.e. spacing greater than well diameter). The spatial properties found in the lineament trends (Figures 4 and 5) may therefore have led to an undersampling of the NE-SW and NW-SE lineament sets and oversampling of the E-W ones.

(iii) The Tchirezrine II deposit is a reservoir with heterogeneous petrophysical properties (i.e. porosity spanning over 26% and permeability over 5 orders of magnitude). The positive correlation observed between the $P10$ structures density and permeability suggests a dominant influence of deformation structures on this petrophysical properties scattering. Although there is no clear correlation between grain size and porosity-permeability relationship, a part of the scatter can be related to grain size and sorting (Figure 10a). We however have not enough data to precise these effects. It is worth mentioning that these studied petrophysical properties estimated from logging tools potentially contain errors due to data treatment. For the sonic porosity estimation in water saturated media, using the Wyllie et al. (1956) equation (eq.1), the rock matrix is supposed to be homogeneously made of quartz (i.e. the fraction of matrix wave velocity is directly correlated to the fraction of fluids within the porosity). However, the Tchirezrine II sandstones contain a variable fraction of clays and analcimes (Billon, 2014; Mamane Mamadou et al., 2022), which are less dense compare to quartz and lead to an overestimation of the porosity value. Concerning the NMR permeability, note that the calculation is based on an empirical equation (eq.2) in which the relaxation time ($T_2$) is considered to be only related to matrix pore space. The impact of open fractures on the $T_2$ distribution remains poorly understood in the literature (see Golsanami et al. (2016) for a review on the application of the NMR technology for investigating fractures). It is however well known that the presence of open fractures increases the estimated permeability, but also that the presence of magnetic minerals reduce the estimated permeability (Jácomo et al., 2018, 2020), but in ranges much lower that the main effect of fracture porosity. The fact that there is a clear correlation between permeability and $P10$ fracture density suggests that these effects are probably minor.

(iv) The hydrogeological testing implemented in a zone of E-W trending structures reveals a substantial anisotropy of flow, with an influence that is 10 times greater in E-W direction compared to the N-S direction (Figure 9a). The tracings show a significant contrast in brine restitution, highlighting fluid channelling in the E-W direction as well as limited flow in the N-S direction. Also note that the maximum flow is observed in a direction (E-W) parallel to the main structural pattern observed in OBI at this site. However, the real anisotropy of flow cannot be more precisely defined since only two orthogonal monitoring piezometers were deployed during this preliminary aquifer testing. We must also note that other orientations of deformation structures, such as N-S or N070°E structures, detected from satellite image analysis (Figure 3-Z3), may lead to permeability anisotropy different from the results shown for IMOU_2527_2. These results, obtained from only one site, containing heterogeneously distributed E-W structures (see (ii)), probably have specific structural and sedimentological configuration and cannot be generalized to the entire Tchirezrine II reservoir. It however reveals the strong influence of the E-W trending structures on fluid flow where they are present.

These different results reveal that the Tchirezrine II is a heterogeneous reservoir locally controlled by E-W deformation structures of variable spatial distribution and typology. The link between the reservoir anisotropy detected during the hydrogeological test and the structural network is discussed below.



### 5.2 Interpretations of lineaments sets

The studied lineaments probably correspond to different types of structures such as faults, Mode I fractures and deformation bands. The two main sets of detected lineaments, i.e. NE-SW and NE-SW trending sets, are characterized by a sub-orthogonal organization (Figure 3). Their spacing coefficient of variation ($C_v$<1) reveals an overall regularly spaced pattern (Gillespie et al., 2001; Odling et al., 1999; Strijker et al., 2012; Watkins et al., 2015). The organization of such sets is variable from one sampling window to another but evolves similarly for both sets. Their length distribution clearly shows that these lineaments are scale dependent (negative exponential law). All these observations are quite consistent with pattern of Mode I fractures mechanically restricted to layers of variable thickness, so called joint sets (Bai et al., 2000; Bai and Pollard, 2000; Hu and Evans, 1989; Odling et al., 1999; Rives et al., 1992; Soliva et al., 2006; Soliva and Schultz, 2008). The mean spacing obtained for these sets and the relationship between joint spacing and thickness of mechanical layers described in the literature (Gillespie et al., 2001; Ji et al., 2021; Strijker et al., 2012) suggest that mechanical units of several meters thickness are affected by these sets. This is also consistent with the description of the Tchirezrine II deposit as fluvial sequences ranging in thickness from a few decimetres to several meters (Orano, Emmanuelle Chanvry personal communication, 2023). As mentioned before, the relatively large spacing observed, compared to data from literature, is probably due to the truncation bias for lineament detection.

Faults are obviously also present in the detected lineaments (Figures 2c, d, and e). E-W lineaments have (1) coefficient of variation of spacing $C_v$≥1 (clustered spatial distribution), (2) size distribution that are more censored than the other sets (section 4.1.2), and (3) maximum lineament length significantly larger than the two other sets (L ≥ 150 m), which is inconsistent with both Mode I fractures and deformation band length (Schultz et al., 2008). Although a component of this set is spatially distributed (negative exponential size distribution, more consistent with fractures or deformation bands), these observations suggests that this E-W set is less scale dependent (more scale invariant) than the NE-SW and NE-SW sets, and therefore more spatially heterogeneous. Faults are generally described in the literature as clustered systems having power-law size distribution, i.e. scale invariant (e.g. Watterson et al., 1996; Yielding et al., 1996), suggesting that this E-W set corresponds better to a fault system. The fact that the E-W faults are more censored than the two other sets can explain a part of the curvature on this graph and an apparent exponential distribution. In addition, it is known that fault systems can show hybrid behaviour (between exponential and power law size distribution) in layered host rock conditions (e.g. Soliva and Schultz, 2008). This behaviour might be relevant to fluviatile deposition context in which stratigraphy is heterogeneous, as described for the studied Tchirezrine II sandstone series. Eventually, it is worth considering that mechanical stratigraphy can have more influence on the growth of small-scale structures than large ones. By the way, small scale structures, such as Mode I fracture and deformation bands (e.g. Fossen, 2016), are inherent to all fault DZ (Schueller et al., 2013) and background deformation (Mayolle et al., 2023). If this is true in our case, a part of the exponential distribution observed, especially the curvature on the graph for small lineament size, could be a real geometrical property of this E-W set. Also note that another lineament set oriented N°070E, much less present in the studied data compared to the other lineament sets, is also described as faults (e.g. Sani et al. (2020), Figures 2a and e, Figure 3) and observed together with E-W lineaments at the basin-scale.

### 5.3 Petrophysical properties and deformation structures

A complex interplay of geological parameters controls the petrophysical properties of NFR (Narr et al., 2006; Nelson, 2001) (Figure 10a). Both matrix and deformation structures impact the porosity and permeability values measured by logging tools (Figure 10b). As discussed above, the Tchirezrine II reservoir exhibits a wide variety of structural features, including Mode I fractures, cataclastic deformation bands and faults that are known to have a contrasting impact on the reservoir's porosity and permeability properties:



(i) Dilatant fractures, when open, are characterized by their ability to slightly increase porosity (typically below 0.5%; Nelson, 2001), and significantly enhance permeability (Sibson, 1996). Open fractures may form preferential pathways for fluid flow and may enhance the overall drainage of the rock matrix (Nelson, 2001; Warren and Root, 1963; Watkins et al., 2018). Where fractures are prevalent along the well, the dataset shows high permeability for relatively low porosity whatever the matrix grain size (Figure 8d). Figure 10c, using data only from FSs of the Figure 8d, reveals the same dominant trend

controlled by fracture density. Note that cemented and fine sandstones with observed fractures (yellow and red clusters) show relatively low porosity (<20%), which constitutes favourable condition for Mode I fracture formation and permeability enhancement (e.g. Wong et al., 1997; Nelson, 2001), and explains the inconsistency of porosity decrease as a function of fracture density (Figure 10c). However, as mentioned before, borehole observations are not relevant to analyse any correlation between high dipping structure density and lithology, especially when boreholes are nearly vertical.

(ii) Cataclastic deformation bands induce a grain size reduction by crushing (comminution) and have a contrasting impact on reservoir's porosity and permeability. These structures can form in various sandstone granulometry since porosity is higher than 15-20% during their formation, and tend to slightly reduce the overall porosity and decrease the permeability of sandstone reservoirs, but with different intensity as a function of various geological factors (e.g. initial porosity, stress regime, burial depth and grain size of the host material, Ballas et al., 2015). In addition to cataclasis, cementation, both

in Mode I fractures, dilation bands and cataclastic bands, can significantly reduce reservoir and fault zone permeability and generate seals or transient barriers to fluid flow. As mentioned for mode I fractures, borehole observations are not relevant to analyse any correlation between their density and lithology.

      (iii) Faults can significantly affect reservoir permeability by two contrasting ways: increasing or reducing permeability, both in the fractured DZ and/or the fault core (Caine et al., 1996). The nature of deformation process, i.e.

disaggregation cataclasis, clay smearing or cementation, and their in-situ stress conditions will govern the capacity of these zones to form efficient permeable or seal structures (e.g. Barton et al., 1995; Yielding et al., 1997; Aydin, 2000; Fisher and Knipe, 2001; Philit et al., 2019).



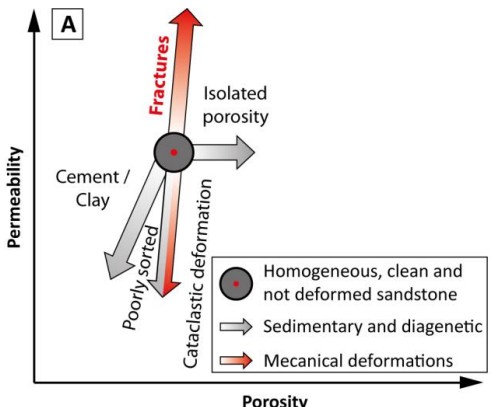

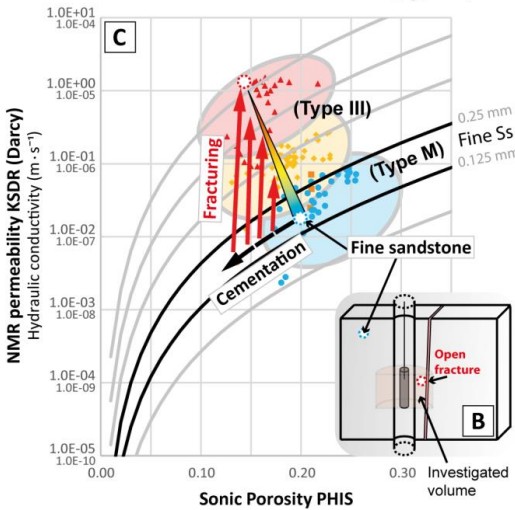

**Figure 10: Porosity-permeability relationship and triggering processes in porous and fractured Imouraren reservoir. (A) Schematic graph showing the impact of various processes on the evolution of the porosity-permeability relationships. Modified from Schön (2015) after Lucia (2007) and Nelson (2005). (B) Scheme showing the potential influence of a fracture in the investigated volume of NMR logging tool. (C) Application of the conceptual model to the porosity-permeability data from only the fine-grained sandstones of the Tchirezrine II, sorted by fractures _P10_ density (see Figure 8d for legend). Arrows represent the evolution of the porosity-permeability relationship. Red arrows are for theoretical fracturing process, the black arrow is for theoretical cementation process, and the rainbow arrow illustrates the resulting evolution in the Imouraren reservoir.**

Since the Tchirezrine II reservoir has 19% of average porosity, and that fracture porosity in NFR is <1% (Nelson, 2001), it is obvious that a large part of the porosity comes from the matrix. In places with low _P10_ density such as borehole data IMOU_1384_2 (Figures 7d, 8d and 10c), the reservoir exhibits Type M behaviour following Nelson (2001) classification, i.e. matrix hosts 100% of reservoir porosity and permeability. In case of higher _P10_ density (i.e. IMOU_0236_4, Figures 7d, 8d and 10c), the reservoir exhibits Type III behaviour following Nelson (2001) classification, where porosity is ~100% matrix and permeability is significantly enhanced by open fractures. In places of the reservoir where the presence of deformation bands or sealed fractures have been identified, the reservoir is more likely to correspond to Type IV, in which porosity is 100% matrix and deformation structures reduce the porosity and permeability of the reservoir. It is worth mentioning that lateral and vertical variation of sandstone facies can occur (i.e. porosity change), and may lead to transition zones with the presence of both cataclastic deformation bands and open fractures (Liu et al., 2021). Considering the scale of ISR production cells, different




NFR types then coexist in the Tchirezrine II reservoir consistently with the variabilities of deformation features detected in this reservoir and the heterogeneities of the host sandstone unit.

### 5.4 Reservoir anisotropy and deformation patterns

In this section, we correlate the structural, petrophysical and hydrogeological analysis and interpretations to constrain the reservoir behaviour and discuss the ISR application in NFRs context.

In the configuration of the hydrogeological test (Figure 11a), the approximate 1/10 ratio of E-W extension of the drawdown cone reveals an anisotropic behaviour of permeability in the Tchirezrine II reservoir, at least at a decametric scale and at a specific site (Figure 9). Tracing in the E-W direction reveals a channelled flow with a high and rapid restitution rate, whereas

tracing in the N-S direction reveals dispersive flow with slow and low recovery (Figure 11b). In this case, anisotropy cannot be linked to a specific sedimentary architecture, since fluvial sandstone bodies are globally oriented N-S (Valsardieu, 1971). However, the lateral connectivity and continuity of the sandstone fluvial bodies remain important since sandstones facies and fluvial bodies are controlling the deformation structure typology and organization (i.e. spacing, vertical and lateral extension). This E-W anisotropy seems mainly controlled by the Mode I fractures and cataclastic deformation bands, both with E-W

azimuth as detected from OBI approach (Figure 7). These deformations are consistent with the orientation and size distribution of the E-W set of lineaments and show large density close to N070°E faults (see section 4.2, Figures 1d and 7d). The anisotropy of the Tchirezrine II reservoir appears then mainly controlled by the N070°E and E-W fault networks (Figures 2a and b) and related damage zones. The crush breccias that are sometimes composing the fault cores (Figure 7c) are mainly observed with clay infilling which probably form E-W seals and also guide fluid flow in this same direction. In fault DZ or far from faults,

open fractures must enhance drainage in the reservoir in the E-W direction, while sealed fractures and cataclastic deformation bands should minor N-S permeability (Figure 11b, e.g. Ballas et al., 2015; Bisdom et al., 2016). The vertical occurrence of brittle (fault, Mode I fractures) vs. ductile structures (cataclastic deformation bands) mainly depends on stress conditions applied and petrophysical properties of the host sandstone (see section 5.3 and references therein), and therefore on the sedimentary architecture of the reservoir (Figure 11a). This high permeability anisotropy in the E-W direction is specific to

areas of high density of E-W deformation structures and should therefore be lower far from these structures (Figure 11c). Note that two other tracer tests were performed on site IMOU_1228_2 but are not presented in this study as their configurations are not relevant for a proper comparison (i.e. different tracer directions to those presented in this study). However, we observe faster N-S brine restitution than at IMOU_2527_2 site, with possible structural drainage. More generally, far from clustered fracture zones we expect structurally more homogeneous behaviour and also behaviour controlled by sedimentary architecture.

For ISR mining purposes, the overall structural network and sub-networks of the sandstone bodies are critical, as traditional ISR mining cells (i.e. a set of injection wells surrounding a recovery well) are decametric in scale (Mudd, 2001). In such a wells configuration, open fractures with penalizing orientation can link the injection wells to the recovery one and act as bypass, preventing the leaching solution from accessing the main ore volume located in the matrix far from the bypass (Figures 9b and 11c). Conversely, cataclastic deformation bands, which generally are baffles to fluid flow, will prevent the leaching

solution from fully accessing the matrix porosity. To inhibit the impact of bypass structures, an ISR well pattern perpendicular to the main permeability direction could be a coherent solution to minimize channelled flow in open fractures (Odling et al., 2004) and force solution through cataclastic bands (Figure 11c - an N-S well scheme for U. Imouraren ore). This scheme would significantly increase the spreading and travel time of the leaching solution, which is better for the kinetics of the leaching reaction than a shorter, channelled path. This can be illustrated by the E-W well pattern shown in Figure 11c, where the leach

solution is channelled directly to the production well, minimizing the volume of accessible matrix. Whatever the considered well pattern, the impact of faults and fractures on top and basal seals integrity is also decisive to prevent any potential leaks of



the leaching-solution. These proposed patterns are based on only one in-situ hydrogeological tests. Further tests could be carried out to gain a better understanding of the lateral anisotropy of permeability and the impact of mechanical-stratigraphic partitioning on fluid flow at the scale of an ISR production cell.


**Figure 11: Schematic view of the Tchirezrine II NFR at the Imouraren site. (A) 3D schematic bloc diagram summarizing the pattern of deformation structures of the southern part of the Imouraren site. (B) Sketches illustrating the potential impact of deformation structures on the fluid flow for different mechanical-behaviour contexts. (C) Two different wells patterns for ISR mining: (N-S) where injectors and producer wells are perpendicular to the main permeability direction minimizing channelled flow between wells,**
**and (E-W) where injectors and producer wells are subject to channelled flow. Scales are not representative.**



**Conclusion**

In this study, we present an original integrated approach, coupling characterisation of fracture networks from satellite image analysis, description of deformation typology from core and OBI, petrophysical properties from well logging data, and aquifer testing to characterize a NFR in heterogeneous fluvial-sandstone sequence of the Imouraren Uranium deposit (Niger). Three main points can be highlighted from our results:

(1) Sub-orthogonal joint sets (NE-SW and NW-SE) are scale dependent in size distribution, widely and homogeneously distributed in the reservoir, but poorly observed in borehole. Conversely E-W deformation structures (faults, Mode I fractures and cataclastic deformation bands), with better scale-invariant size distribution and therefore having spatially heterogeneous distribution, are frequently observed in some boreholes. They appear to be clustered around faults and provide strong heterogeneity in the structural framework.

(2) The porosity of the Tchirezrine II reservoir appears to be mainly host by the sandstone matrix. In tighter parts of the reservoir (low-porosity fine-grained units), the permeability is enhanced by open mode I fractures, as a function of their $P10$ density (reservoir Type III). Conversely, in coarse-grained and porous sandstone the permeability is relatively low (i.e. compared to theorical Kozeny-Carman trend), potentially reduced by the presence of cataclastic deformation bands (reservoir Type IV). The different typologies of deformation structures reveal the impact of the initial porosity of the host rock on deformation mechanisms and, ultimately, on the diversity of reservoir properties.

(3) Salt tracing and drawdown tests at IMOU_2527_2 site reveal a significant higher fluid flow E-W than N-S, probably related to the observed E-W trending deformation structures. These structures must impact water flow with preferential fluid pathways in open mode I fractures and/or guides to flows along transient seals. This site reveals that fluid flow can be highly anisotropic (ratio 1/10 of the drawdown cone) into zones of E-W trending structures, which are heterogeneously distributed in the reservoir. In these areas, strategies can be implemented to limit well-to-well bypass and globally improve Uranium ISR mining in NFRs by integrating the general fracture network, its typology and permeability anisotropy. Far from these E-W structures, permeability anisotropy should be less pronounced.

*Author Contribution Statement.* **Maxime Jamet**: Conceptualization, Methodology, Lineament data acquisition, data QAQC, Formal analysis for Lineament and Wells data, Writing, Original draft preparation. **Gregory Ballas**: Conceptualization, Formal analysis, Writing-Reviewing and Editing. **Roger Soliva**: Supervision, Conceptualization, Lineaments Methodology and Formal analysis, Writing-Reviewing and Editing. **Olivier Gerbeaud**: Orano project administrator, Structural data acquisition form OBI, Reviewing and Editing. **Thierry Lefebvre**: Data acquisition, data QAQC, Formal analysis, Interpretation and Reviewing for hydrogeological testing. **Christine Leredde**: Conceptualization, Formal analysis, Reviewing. **Didier Loggia**: Conceptualization, Formal analysis, Reviewing.

*Competing interest.* The contact author has declared that none of the authors has any competing interests.

*Acknowledgements.* We would like to thank Orano Mining for its financial support (grant no. 211783-02) and for providing valuable on-site data for this study. We would also like to thank R. Mieszkalski, Y. Bensedik and G. Dufréchou from Orano for their help in processing the log data and for their advice. We would also like to thank E. Chanvry of Orano for her help in understanding the sedimentary context.

*Data availability.* The datasets in this article are available on request only.



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
