# Peer review of "Naturally fractured reservoir characterisation in heterogeneous sandstones: insight for Uranium In Situ Recovery (Imouraren, Niger)"

_EGUsphere, 2024_

## Referee Comment (RC1)

Review EGUsphere
Title: Naturally fractured reservoir characterization in heterogeneous sandstones: insight for Uranium In Situ Recovery (Imouraren, Niger)
Author(s): Maxime Jamet et al.
MS No.: egusphere-2024-435
MS type: Research article

**Evaluation of the overall quality of the preprint ("general comments")**

This is a very concise and well-presented research work integrating multifaceted analysis technics of reservoirs and fracture networks applied for the very first time to ISR of uranium resources.

The problem is well addressed in the introduction and well developed throughout the manuscript. Despite the fact that the reader has to read 8 pages before arriving at the results, the results are very rich and with a lot of detail (which sometimes makes it difficult to read).

After a very detailed quantitative and typological fracture analysis, the manuscript emphasizes the role of various types of tectonic structures on the anisotropy of the permeability of the Imouraren uranium reservoir controlling favorable versus unfavorable fluid flow pathways, and makes suggestions for the future ISR infrastructures management. This is a very nice case of applied research.

In addition, the authors propose à nice model of the 3D architecture of the reservoir coherent with all the new data collected. This 3D model is from far the best one has been produced up to now of the Imouraren uranium deposit.

For all these reasons, this article appears to me of high interest for the whole geoscientist and industrial community, and particularly for the ISR actors in uranium resources.

In my opinion, the manuscript just requires some corrections/modifications of second order before final publication.

**Individual scientific questions/issues ("specific comments")**

Is "attenuation by fracture networks" what is searched with ISR production?

Is "Mode I" still relevant in modern fracture analysis knowing that in 3D all three "modes" are represented in the same fracture?

I am surprised to do not see any description of "horizontal" compressive structures. These are mentioned and illustrate in a previous Areva internal report from oriented bore-holes data and supported with field pictures. These "horizontal" set is obviously difficult to document from satellite images but should be observable on new drill-cores and OBI data. I wonder what can be the influence (favourable or unfavourable) of such fault set in the permeability behaviour, the fluid flow and finally in the recommended ISR infrastructure… (as said by the authors in line 651 …the impact of faults and fractures on top and basal seals integrity…).

If I have correctly understood, the 4 fracture sets identified at the basin scale (ENE-SWS, ESE-WNW, N-S and NNE-SSW; section 4.1.1) are not the same as the 3 main fracture sets defined from circle area sampling from satellite images (NE-SW, NW-SE and E-W; sections 3.1.2 and 5.1), and retained for the discussion and conclusions. Why don't they match? Why the shift between those sets? What happens with the "N-S Arlit fault type" sets as warned in lines 523-525)? Additionally to these

questions, I note that the basin scale lineaments are not sub-orthogonal (set 1 vs set 2, pag 8). Their shift to NE-SW and NW-SE makes them sub-orthogonal. It looks like a simplification of regional sets orientation to make them sub-orthogonal at the deposit scale? Isn't it a little bit abusive?

**List of technical corrections ("technical corrections": typing errors, etc.)**

Abstract
Line 12. Why "complex" reservoir? Better "heterogeneous"…
Line 18. Mode I fractures. Is this still relevant in modern fracture analysis?
Line 18.  Is "brecciated" needed?
Lines 26-27. Is "attenuation" what is searched with ISR production?

Keywords
You could add "ISR"

Introduction
Line 35. "…especially the transition to low-carbon energies (Evans et al., 2009)" not needed
Line 60. "…following the brittle-ductile transition of such porous rocks…" This is confusing here when talking about sedimentary rocks! Needs precision or better remove it.
Fig. 1b. Add in the legend the meaning of the arrows indicating N120E, N070E and N030E (fault sets)
Fig. 1c. Strange to publish in 2024 à cross section with vertical faults!!!

Material and Methods
Line 136. "…the size of these circular sample surfaces is of the same order of dimensions as a set of ISR cells". I like this.
Line 228. Are these two piezometers indicated in Fig. 1D? In my paper copy the quality of the image is not enough to check for these two piezometers.

Results
Please review Figure 2 information in relation with text from lines 245 through 262.
-    Line 251. Figure 2d (e?)
-    Line 252. d? d is the figure cited for the set N060 (line 247). Please check.
-    Line 255. Figure 2f shows strata not fractures!!! Figure 2f. Please draw a fold axis, or indicate fold limbs dip (strata without any dip information = no meaning).
-    Line 259. Where is Madaouela in Fig. 2a?
Lines 272-273. "These data were used to characterize the structural organization of lineament networks affecting the Tchirezrine II reservoir at the scale of ISR project, i.e. ranging from meter to hundred meters scales" Not needed, already said in methodology. Deleting other sentences like this one could help to reduce the length and repetitiveness of the manuscript.
Line 299. "…we were unable to find a specific value (i.e. which is lower 300 than censored ones)". Not clear why… Maybe you can add complementary information.
Line 367. "… generally clays or oxidized products…". Please explain how you identify such products from OBI, or specify this is done from drill-core direct observation (it is confusing here because you start de paragraph saying from OBI, line 364).
Lines 410 through 435 are a little bit indigestible…

Discussion
Line 478-481. (NE-SW and NW-SE)….  Maybe better ENE-WSW and WNW-ESE to summarize sets 1 and 2 (page 8)… which are not really sub-orthogonal!!!
See also "specific comments".

Conclusions

Nice conclusions!!!

Line 666. Conclusion 1. I still have my doubts about "sub-orthogonal" sets... See specific comments.

Figures

Figure 1. Text in D is unreadable.

Figure 2. Please review information in relation with text from lines 245 through 262.

Figure 10. Arrange horizontaly.

Figure 11. What does "Imola" mean? Why strata traces are so irregular? It isn't nice...

---

## Referee Comment (RC3)

[referee-annotated manuscript omitted]

---

## Author Response (AR1)

**Responses to RC1: Anonymous Referee #1**

Dear Anonymous Referee,

We would like to thank the anonymous referee for his analysis of the manuscript and his relevant view of the industrial challenges addressed in this article. We have modified and corrected the manuscript according to the feedback and suggestions we received.

You will find below the detailed responses to your comments as well as the corrections/modifications that have been made in the manuscript. We hope that these new elements have significantly improved the manuscript and will enable this work to be published in Solid Earth.

Kind regards,

Maxime Jamet

*Evaluation of the overall quality of the preprint ("general comments")*

*This is a very concise and well-presented research work integrating multifaceted analysis technics of reservoirs and fracture networks applied for the very first time to ISR of uranium resources.*

*The problem is well addressed in the introduction and well developed throughout the manuscript. Despite the fact that the reader has to read 8 pages before arriving at the results, the results are very rich and with a lot of detail (which sometimes makes it difficult to read).*

*After a very detailed quantitative and typological fracture analysis, the manuscript emphasizes the role of various types of tectonic structures on the anisotropy of the permeability of the Imouraren uranium reservoir controlling favorable versus unfavorable fluid flow pathways and makes suggestions for the future ISR infrastructures management. This is a very nice case of applied research.*

*In addition, the authors propose à nice model of the 3D architecture of the reservoir coherent with all the new data collected. This 3D model is from far the best one has been produced up to now of the Imouraren uranium deposit.*

*For all these reasons, this article appears to me of high interest for the whole geoscientist and industrial community, and particularly for the ISR actors in uranium resources.*

*In my opinion, the manuscript just requires some corrections/modifications of second order before final publication.*

*Individual scientific questions/issues ("specific comments")*

*Is "attenuation by fracture networks" what is searched with ISR production?*

As far as we know, mineralization is mainly located in the porosity of the sandstone matrix. Under dynamic conditions in fracture/matrix porosity media, leaching fluid mainly flows through preferential circulation paths such as open fractures, which can in an end member case totally bypass the matrix. One of the challenges of ISR feasibility is to know whether the leaching solution will be able to reach the largest volume of porosity matrix, which contains the mineralization.

*"Mode I" still relevant in modern fracture analysis knowing that in 3D all three "modes" are represented in the same fracture?*

Yes. However, as the second reviewer suggested changing this terminology and following the terms published in Peacock et al. (2016), we have changed this term to 'extensional fracture' throughout the manuscript.

*I am surprised to do not see any description of "horizontal" compressive structures. These are mentioned and illustrate in a previous Areva internal report from oriented bore-holes data and supported with field pictures. These "horizontal" set is obviously difficult to document from satellite images but should be*

*observable on new drill-cores and OBI data. I wonder what can be the influence (favourable or unfavourable) of such fault set in the permeability behaviour, the fluid flow and finally in the recommended ISR infrastructure… (as said by the authors in line 651 …the impact of faults and fractures on top and basal seals integrity…).*

In this study, we only consider structures that are geometrically constrained by the OBI in order to compare them with satellite data. As described in section 4.2, we mainly observed sub-vertical structures, meaning that (1) these horizontal compressive structures are no present in the studied boreholes, or (2) they are below the resolution of optical borehole images. Since there are no more drill cores of the data mentioned in the AREVA report and that these observations where made in the Imfout area (4 km North of the study area), if any, we cannot analyse these structures. A study of the deformation structures at a lower scale is seen as a prospect for a better characterization of the Imouraren structural framework. Since we do not have any indication on presence, the typology and the spatial distribution of these structures we believe irrelevant to discuss their impact on fluid flow in the reservoir.

*If I have correctly understood, the 4 fracture sets identified at the basin scale (ENE-WSW, ESE-WNW, N-S and NNE-SSW; section 4.1.1) are not the same as the 3 main fracture sets defined from circle area sampling from satellite images (NE-SW, NW-SE and E-W; sections 3.1.2 and 5.1), and retained for the discussion and conclusions.*

*Why don't they match? Why the shift between those sets?*

The nature of the structures observed changes with the scale of observation, at the scale of the basin we only observe faults, where at the reservoir scale, we observe smaller-scale structures.

*What happens with the "N-S Arlit fault type" sets as warned in lines 523-525)?*

There are very few N-S structures in the study area. That's why we did not analysed and discuss them in the paper since there is much more prominent structures and subjects to consider in the paper. We believe they are part of the N-S Arlit fault damage network, and the large distance of the outcrops from the Arlit fault can explain their low amount consistently with Mayolle et al. (2023), showing the spatial distribution of damage in a full fault network. This is now explained in the discussion in lines 537-538.

*Additionally to these questions, I note that the basin scale lineaments are not sub-orthogonal (set 1 vs set 2, pag 8). Their shift to NE-SW and NW-SE makes them sub-orthogonal. It looks like a simplification of regional sets orientation to make them sub-orthogonal at the deposit scale? Isn't it a little bit abusive?*

We do not understand this comment. We think the referee confused the N060-080°E (ENE-WSW) set from basin scale interpretation with the N030-060°E (NE-SW) from circle sampling analyses, which we actually consider as sub-orthogonal to the N110-140°E (NW-SE) set, also from circle analysis. This point has been clarified in the discussion section 5.1 and 5.2.

***List of technical corrections ("technical corrections": typing errors, etc.)***

***Abstract***

*Line 12. Why "complex" reservoir? Better "heterogeneous"…*

We have changed it (Line 12)

*Line 18. Mode I fractures. Is this still relevant in modern fracture analysis?*

We have changed this term by "extensional fracture" in the whole manuscript.

*Line 18. Is "brecciated" needed?*

In fact, we have observed both cataclasite and breccia from faults cores (see section 4.2 - (iii)).

*Lines 26-27. Is "attenuation" what is searched with ISR production?*

We have answered this question in the specific comment section. There are also elements of response / justification in section 5.4 (line 657-674)

***Keywords***

*You could add "ISR"*

Done

***Introduction***

*Line 35. "…especially the transition to low-carbon energies (Evans et al., 2009)" not needed*

This quote reviews the current and future uses of geological reservoirs. We think it's a good overview and that this citation is appropriate for highlighting the stakes involved in the use of geological reservoirs.

*Line 60. "…following the brittle-ductile transition of such porous rocks…" This is confusing here when talking about sedimentary rocks! Needs precision or better remove it.*

We have modified this part of the sentence to avoid any confusion: *"following the transition between brittle and cataclastic deformation of such porous rocks".* (Line 62)

*Fig. 1b. Add in the legend the meaning of the arrows indicating N120E, N070E and N030E (fault sets)*

These arrows refer to the different sets of faults described in the 2.1 section. We updated the figure caption to better highlight this. (Line 12-122)

*Fig. 1c. Strange to publish in 2024 à cross section with vertical faults!!!*

This cross-section is constrained in depth by drills but at this time we are not able to better constrain the faults dip.

***Material and Methods***

*Line 136. "…the size of these circular sample surfaces is of the same order of dimensions as a set of ISR cells". I like this.*

We are glad to hear it.

*Line 228. Are these two piezometers indicated in Fig. 1D? In my paper copy the quality of the image is not enough to check for these two piezometers.*

We've updated figure 1 to enlarge sub-figures 1d and c. We have also changed the ID colour of the aquifer test boreholes to blue.

**Results**

*Please review Figure 2 information in relation with text from lines 245 through 262.*

*- Line 251. Figure 2d (e?)*

Done.

*- Line 252. d? d is the figure cited for the set N060 (line 247). Please check.*

We've removed figure 2d reference.

*- Line 255. Figure 2f shows strata not fractures!!! Figure 2f. Please draw a fold axis, or indicate fold limbs dip (strata without any dip information = no meaning).*

We've updated figure 2f drawing the fold axis.

*- Line 259. Where is Madaouela in Fig. 2a?*

The presence of the hydrographic network makes it difficult to observe the Madaouela structure in satellite images, particularly near Imouraren. However, it is visible in sub-figure 2d, located to the north of sub-figure 2a.

*Lines 272-273. "These data were used to characterize the structural organization of lineament networks affecting the Tchirezrine II reservoir at the scale of ISR project, i.e. ranging from meter to hundred meters scales" Not needed, already said in methodology. Deleting other sentences like this one could help to reduce the length and repetitiveness of the manuscript.*

We have removed this sentence.

*Line 299. "…we were unable to find a specific value (i.e. which is lower than censored ones)". Not clear why… Maybe you can add complementary information.*

Sand deposits produce artificially large spacing values that we do not intend to interpret. For the E-W set, there is no censored spacing value that is larger than the largest uncensored spacing. We have changed this part of the sentence to be clearer (Line 311-312), also we moved the previews sentence to the methodology 3.1.2 section (line 172-174).

*Line 367. "… generally clays or oxidized products…". Please explain how you identify such products from OBI, or specify this is done from drill-core direct observation (it is confusing here because you start de paragraph saying from OBI, line 364).*

We have specified these observations are done from drill-cores. (Line 379)

*Lines 410 through 435 are a little bit indigestible…*

We're sorry about that, this section aims to be as complete and succinct as possible, which is difficult given the large volume of data to be described.

**Discussion**

*Line 478-481. (NE-SW and NW-SE)…. Maybe better ENE-WSW and WNW-ESE to summarize sets 1 and 2 (page 8)… which are not really sub-orthogonal!!!*

*See also "specific comments".*

As mentioned before, we refer to second-order lineament sets that are described in the 4.1.2 section. As you can see Figure 3a, b and c. these sets are effectively sub-orthogonal. To clearly state that we've added precisions in the 5.1 and 5.2 section. (line 491-492, 545)

**Conclusions**

*Nice conclusions!!!*

*Line 666. Conclusion 1. I still have my doubts about "sub-orthogonal" sets… See specific comments.*

See previous answer.

**Figures**

*Figure 1. Text in D is unreadable.*

We have upscaled this subfigure (d) making it readable.

*Figure 2. Please review information in relation with text from lines 245 through 262.*

This is done, thank for the comments.

*Figure 10. Arrange horizontaly.*

Done

*Figure 11. What does "Imola" mean? Why strata traces are so irregular? It isn't nice*

Imola is the southern part of the Imouraren deposit, we changed it by *"South Imouraren"*.

This is a conceptual model of the structural architecture of the southern part of the Imouraren deposit, without vertical and horizontal scales. It is not intended to give a reliable picture of the reservoir with all its sedimentary complexity. We have opted to represent heterogeneous sandstone channels in order to highlight some of this complexity.

**Responses to RC3: Nikolas Aleksi Ovaskainen Referee #2**

Dear Mr Ovaskainen,

The authors would first like to thank Nikolas Aleksi Ovaskainen for his thorough review. The fair questions and comments raised have helped us to clarify points throughout the sections, and we are grateful for this.

You will find a detailed response to your comments in the following document. To sum up, we have made significant changes to the methodology in order to incorporate greater detail, particularly in the lineament section, but also in the drilling and hydrogeological testing sections. We have also made the whole manuscript more fluent by correcting the syntax and using consistent terminology. The different corrections and references proposed were also considered.

We hope that these new elements have consistently improved the manuscript and will enable this work to be published in Solid Earth.

Kind regards,

Maxime Jamet

*The manuscript by Jamet et al. uses a combination of methods to characterize a potential reservoir where Uranium In Situ Recovery might be done. The study uses methods that are well fit for purpose for this task and is based on an extensive dataset of e.g. lineaments which is introduced well in numerous figures. The introduction is well structured where first a review of prior studies is presented, then the authors make successful claims on why their study is required and then present the agenda of how their study will answer this need. The topic is an interesting one and fits well within the scode of Solid Earth through a focus on structural geology and rock deformation applied in a hydrogeological context.*

*However, the methodology section requires a major extension and the authors need to make sure all presented methods are introduced there, with sufficient detail and with appropriate citation, rather than in other sections of the manuscript. I would especially like for the authors to look at previous publications from Solid Earth with similar topics for reference and/or citations (See References).*

The authors have significantly enriched the methodology section by adding details and references to previous studies, in particular those proposed. The detailed explanation of the revision made is exposed below in the point per point reply to the referee comment.

*Furthermore, the manuscript claims to be a multi-scale study but I find the analysis of "basin scale" lineaments lacking. A multi-scale study, in this context and in my opinion, should use comparable methods in different scales of observation and then compare the results consistently. Now the study mostly focuses on integrating results from different study methods rather than comparing results from a single/comparable method applied in multiple scales.*

With regard to the term "multi-scale" used, we argue that the different scales of observation (satellite ilages, bore hole) and the different methods used in the study allow us to carry out both "multi-scale" and "integrated" analysis in the discussion (section 5). We believe that the term "multi-scale analysis" in science is not reserved to the use of the same methodology at different scales.

The lack of lineament analysis at the basin scale is justified by a) the fact that a large area at this scale is represented by old rocks and past deformations are recorded in the basin before the Tchirezrine II deposit, b) the large amount of recent sedimentary deposits out of the study area makes difficult observations at the basin scale. We believe this is not useful to explain in the paper since this is well visible on the satellite images, but if the editor think we must explain it, we will do it.

*I also suggest that referring to the digitized features from target areas Z1-Z4 as "fractures" rather than "lineaments" could be more appropriate, and strengthen the multi-scale claim, as the features in the*

*satellite images can probably be interpreted as real fractures whereas "basin scale" lineaments can not be so confidently identified as bedrock structures.*

We prefer to systematically use the term "lineament" in the manuscript before any interpretation about the nature of the different sets. As described in section 5.2, we interpret these different sets as being of different natures (NE-SW and NW-SE vs E-W).

*You should also specifically mention, when introducing fracture network characterization methods such as spacing analysis, which values from the analysis are scale-independent and which are not (E.g. is power-law exponent scale-independent? Is P21 scale-independent?).*

Since all the power-law exponents found in our analysis are close to 2, which defines scale invariance (e.g. Berkowitz and Adler, 1998), then the type of law found (power vs exponential) between two variables expressed as two physical parameters (such as frequency, lineament length or spacing) defines the scale invariance or dependence of these parameters. This is now mentioned in the manuscript in line 185 of the methodology section. Note that constant parameters in the law, as the proportional constant, or the proportional constant within the exponent of an exponential law, does not contribute to differentiate scale dependence and scale invariance. Also note that this comment appears irrelevant concerning P21 density since we do not analyse thoroughly its spatial distribution to be able to analyse its scaling.

*In terms of manuscript text quality and language, the quality of text seems to decrease a bit in the methodology section in comparison to e.g. the introduction and the discussion. The method section should be critically reviewed for all typos and grammar errors. Furthermore, you should make sure you use either American or British English. Currently e.g. both "characterization" and "characterisation" are found in text. In terms of terminology, I think you should critically review terms related to fractures. You use e.g. "mode I fractures", "joints", "lineaments", "faults", "fractures" and "deformation structures" intermixed and sometimes in inappropriate context. Try to simplify the terminology (mode I fractures = joints?) and refer to the same features with the same terminology consistently. You should also consistently refer to azimuth sets, they are sometimes joint sets, sometimes fracture sets and sometimes orientation sets. See e.g. Peacock et al., 2016 for a glossary.*

We have made a number of corrections and changes to improve and harmonise the quality of the manuscript. We have also changed the term "mode I fracture" by "extensional fracture" and the term "size distribution" by "length distribution". In addition, we now refer to the glossary in Peacock et al. (2016). (line 215)

*Please check the author/submission guidelines to make sure your figure labels and captions are consistent ("Labels of panels must be included with brackets around letters being lower case (e.g. (a), (b), etc.).").*

We hope that we have made the necessary changes to better match the review's editorial requirements.

*For the stated reasons, specific comments below and suggested technical corrections, the manuscript requires a major revision. Most importantly, the methodology section must be extended and appropriate citations should be used there.*

***Responses to specific comments by line(s):***

*12-13: I suggest breaking the sentence into two: "in North Niger. The characterization is crucial …"*

Ok done (line 13)

*16-17: a) Why not use "scale-dependent" and "scale-independent" i.e. antonyms to highlight the contrasting observations? b) I also do not immediately understand how is scale-dependency or lack of it related to spatial homogeneity? I suggest either explaining this better in the abstract or rewording a bit if suggesting this relation is not your intent. E.g. "Lineament analysis reveals scale-dependent patterns in spatially homogeneous joint networks whereas, in contrast, in spatially heterogeneous fault networks the patterns are nearly scale-independent."*

a) As far as we know, the terms "scale-dependent" and "scale-invariant" are the most appropriate and used for these concepts. b) This comment has been done reading the abstract where we are not allowed to put references. The link between fault/fracture length and spatial distribution has been thoroughly

studied by Soliva et al., 2006 and Soliva and Schultz (2008). These papers are mentioned in the manuscript where relevant. Please tell us if you see the need of more explanation or references.

*19: Include what is the observation based on (shortly). E.g. "Based on ... data/results, the Tchirezrine II ...". Or if the observed/assumed sedimentary context is the basis, move it to the start of the sentence.*

We have modified it (line 19)

*23: What does "this heterogeneity" refer to? To me it seems redundant and the end could be removed: "... a strong permeability anisotropy."*

This heterogeneity corresponds to the zone of E-W trending deformation structures, but we have removed this part of the sentence (line 23)

*33-37: I suggest trying to find a few more references to earlier works in relation to e.g. oil and carbon extraction and other applications seeing as e.g. research on hydrocarbon extraction from reservoirs has a long history. If possible, the references should try to reflect that history.*

We have added several references to this sentence (line 34).

*45-48: I find this sentence a bit too long. I suggest breaking into parts. E.g. "However, the application of such classification remains a challenge such as in high porosity matrix reservoirs and within polyphased tectonic settings. The multiple tectonic phases and associated processes in a such a setting can make the deformation structures more, or less, favourable or penalizing for petrophysical properties". Please also clarify what you mean by "penalizing for petrophysical properties".*

We have modified the sentence to make it clearer (line 45-48). We were referring to the ability of deformation structures to drain or compartmentalize the reservoir.

*55-73: The justification/claim for the need for this research is stated in this section. I would however try to reword and shorten this section a bit to better highlight the main factors. You could use e.g. a), b), c), etc. labeling to mark them clearly for the reader. Otherwise the justification is somewhat hidden in the quantity of text here.*

We have shortened this paragraph by two sentences merged (line 56-75), and we also have labelled the main factors/reason.

*72-73: I suggest rewording a bit: E.g. "Due to the aforementioned reasons, characterization of the role of such structures on fluid flow in sandstone NFR requires multi-scale and multi-method investigation." "Reasons" here referring to the main factors/reasons a), b), c), etc.*

Done (line 73-75)

*75-78: Reword so the data type is mentioned first, then the area. E.g. "(1) lineaments based on satellite images ...". You should also either list data types or methods, not both in the same list. "hydrogeological tests" -> "hydrogeological test data".*

The study area is now mentioned in the first sentence (line 77). We chose to list the methods because they better describe the multi-method and multi-scale approach of the study. (Line 77-80)

*78: Add few examples of what kind of "hydrogeological tests" (or "hydrogeological test data"; see above) so they are introduced with the same detail as "well log data".*

Done (Line 80)

*84: Mention the main applications of this research here. Something like this: "Based on our findings, future ISR can be better focused…"*

As you suggest, we have added a last sentence concerning future application for ISR management. (line 87-88)

*100-103: As I understand, Figure 1c demonstrates this burial depth, so refer to it here.*

Done. (line 106)

*121-122: These lines list the methods in a different order than in the following subsections and is repetition from the actual Introduction section. Either remove it or reword it so that it explains the workflow in the order presented in the subsections.*

Right, removed.

*127-137: I find the text introducing the different "first-order", "second-order" and "4 different targets" lineaments confusing here. Please clarify clearly what are the (different?) scales of observation and what target areas were used in what scale. The talk of "sets", "*-order" and "targets" intermixed here needs to be made more consistent. I also do not think the concept of "first-order" and "second-order" lineaments can be used without prior explanation. Define the terminology here or in prior text and use it consistently.*

Scales of first and second -order are now define in scale (i.e. 1:3000 and 1:30, respectively). (line 133 and 136)

We have changed thus terms to clearly explain this part by removing "target" and refer to sampling area or eventually to site. (line 130-140)

*136-137: Either introduce the concept of a "ISR cell" in e.g. introduction or move this sentence to the discussion section where it can be explained better.*

We have added *"(i.e. several injection and pumping wells traditionally spaced 10 to 20 m apart)"* in the Introduction section just after the first mention of this concept. (line 82-83)

*138-144: I would assume most of the methodology in e.g. extracting topology parameters follows some prior publication. Due to the similarity word-by-word to e.g. Ovaskainen et al., 2022 it is clear you have taken inspiration from prior publications. Cite it accordingly. My suggestions would include Sanderson and Nixon, 2015; Manzocchi, 2002; Nyberg et al., 2018; Dichiarante et al.,2020; Ovaskainen et al., 2022; Ovaskainen et al., 2023 but most importantly please cite whatever publication you have used yourselves.*

Right, we have taken inspiration in the methodology from several prior publications that deserve to be cited here (not specifically your 2022 paper but it is true that a part of the methodology and several software are similar). We have then added several references at the end of this section. (Line 147-149)

*140: You mention "censored areas like covering sand deposit" and from the text I would understand you are doing an intersection analysis to find lineaments that touch these "censored areas". However, you have not introduced how you have defined these censored areas and I can not find further reference to any intersection analysis. Please clarify.*

As the sand deposits are fairly heterogeneous, we have manually labelled the censored lineaments (i.e. censored by sand coverage). We have modified this sentence to make it clearer. (Line 143-144)

*147-154: I think a reference or two to prior publications you have drawn inspiration would fit here.*

We have added *"(Sanderson and Peacock, 2020)"* on that purpose. (Line 153)

*155-186: Seeing as you introduce many equations in "3.2 Wells data" section, why not also for the fracture network characterization ones such as for P21 and spacing? I think, unless some method is much more crucial that others, all methods that area based on prior works should be introduced with similar detail to keep the methodology section consistent.*

The equations of the section 3.2.3. Geophysical logging and processing are crucial since they explain how reservoir parameters (porosity/permeability) are indirectly calculated/extracted from different geophysical acquisitions. The fracture network characterization is "only" based on geometrical measurements and relationships that, with consistent references such as Dershowitz and Herda (1993), Cox and Lewis (1966) or Odling et al. (1999), not deserve detailed equations.

*170-176: I find this text of low quality. Reword past tense, when referring to prior publications i.e. "was done" and "was described" to e.g. "can be done" and "can be described", respectively. "fracture sets", "fault", "lineament" and "joint" are all used within this small amount of text. Please use consistent terminology when explaining the methods.*

We rephrase parts of this paragraph with your suggestions (line 180-188)

171-172: Please elaborate on the fitting method. Is it e.g. a least-squares fit to log-log plot of lengths (looks like it according to Figure 5)? If it is, this method is somewhat flawed and the caveats should be discussed in the discussion section with references on modern literature of the issue. See e.g. Clauset et al., 2009; Healy et al., 2017. I do not suggest revising your methodology in this manuscript, as the power-law determinations are mostly used as a method of comparison between azimuth sets, just to take this into account when discussing the length distribution results.

We have added specifications on the plots and the fitting method which is a least-squares fit. (line 182-184) We also discuss caveats from least-squares fit in the 5.1 Main results and limitations section. (line 500-502)

175: I would not say "generally described" but that "Joint sets can show scale-dependent characteristics due to mechanical ...".

Yes, done. (Line 186-187)

181-183: The sentence makes it seem like the cut-off of 6 meters is based on literature when it is actually related to the resolution and based on analysis of the length distribution i.e. it is a result. Explain here how you intend to determine the cut-off and then introduce the value in the Results section, not here.

According to us, this cut-off is not a result since it is a resolution-related value and is not based on the length distribution analysis. As mentioned by Bonnet et al. (2001) *"there exist no truly objective methods for determining the lower limit to the scale range over which the length distribution exponent should be determined (truncation length) »* Considering the resolution (0.3 m/pixel) of the images data, we have chosen this 6-meters value in accordance with the cited literature. If we consider a reasonable margin of error of 2 pixels (0.6 m) when sampling the lineaments, this means that with this cut-off, the length values are not truncated greater than 1/10 of their real length. This ratio decreases as the length of the lineaments increases.

188-191: More details about the drilling campaign could be added here and also add a reference to the figure with a map of the well.

However, we are not convinced that adding further details about the drilling campaign is relevant to a better understanding of this part of the study.

191: What is the "Uranium ISR project"? Introduce this project (shortly) in the Introduction section or reword the section about ISR to include the word project so it does not appear out of nowhere.

We have replaced "Uranium ISR project" by "Orano's uranium ISR target". (line 203)

193-195: Please include more detail here. E.g. what "deformation features" and so on.

Features was changed by structures, and we have added example in brackets (fractures, faults, deformation bands…). (Line 205-206)

195: What is defined as "systematically high proportion of clay"?

Orano's internal reports show that very fine sandstones from Imouraren fluviatile sequences are systematically find in presence of high proportion of clay minerals We have added references to Billon (2014) and Mamane Mamadou (2016) to justify this point (line 208). As discussed in section 5.1 clay mineral impact the sonic porosity from Wyllie et al. (1956) equation and do not allow us to properly analyse the impact of deformation structures on porosity-permeability relationship.

205: If I understand correctly, you should elaborate here that by "the azimuthal component" you refer to the strike of the structure extracted from the OBI as "strike" is more easily undersood by a structural geologist.

We have reworded this (line 219).

207-225: 1) Please add more details such as the hardware used and manufacturers to the extent you have the information available. 2) This subsection is nicely succinct but I would try, if possible, to join together some of the paragraphs to make the text flow a bit better. The use of "(i) and (ii)" could be done within a single paragraph rather than as a list. On Line 207, the subsection could be started with a small introductory sentence, e.g. "Geophysical logging data from the vertical boreholes consist of (i) ...".

1) We have added hardware details 2) Following your suggestions, we have grouped the paragraphs in this sub-section, and list the equations after. (line 211-239)

*226-242: 1) Add hardware details of e.g. the piezometer to the extent you have the information available. 2) Add reference to the figure that shows the wells on a map. 3) Try to add references to prior studies that introduced these methods or have used them in similar fashion as in this manuscript.*

1) We have added hardware details. 2) Done. 3) We have added two references with similar methods in this section. (Line 240-257)

*245-262: 1) There seems to be no target area definition for lineaments in the "basin scale". Furthermore, no fracture network characterization results are displayed for these lineaments other than azimuth/orientation analysis. To truly be a multi-scale scale study, I believe you need to expand the analysis of the "basin scale" lineaments. Create a suitable target area, preferably circular, and add details for spacing and length of these lineaments similarly to the target areas Z1-Z4. I do not think you need to do the analysis separately for each azimuth set in the "basin scale", as was done for Z1-Z4, but that is up to you. In the discussion, compare the results, especially the scale-independent variables (power-law exponent), to see how the variables differ between scales. The results of the comparison can probably be used in estimating/discussing the scalability of your other conclusions. 2) I suggest adding a simple plot that contains a visualization of the defined azimuths sets from the two different scales ("basin scale" and Z1-Z4) so they can be easily compared.*

We understand your point of view that, on the basis of basin-scale data, we could make a more thorough analysis of the fault network. There are many reasons for our choice. The first one was that the primary objective of that study basin part, namely a better constrain on the orientations of the basin fault network and a comparison with cross-cut faults within Imouraren deposit. The second one was that, at this scale of investigation, we use a merged orthophotography from different satellites with different angles of observation at different times, this method involving some data heterogeneities. Moreover, at the basin scale the topography is not as flat as at the Z1-Z4 scale, which can easily disturb any potential analysis. The third one was that there are a lot of sand and alluvial cover with heterogeneous distribution (e.g. as the hydrographic network come from the Aïr Massif at East and go down at West, many ENE-WSW clusters of faults are filed with clastic sediments). For these reasons we chose to use this data as a contextualizing structural context and offer the possibility to compare these data with previously published maps on the literature.

Concerning the "multi-scale" used term we argue that the various scales of observation from the different methods we use in the study allow us to perform an integrated analysis as we discuss them all at the same time in section 5.

*287-292: Add reference to the figure where these observations are from.*

Done (Line 302 and 305)

*295: You use "lineament trend" here where before you use "azimuth set". Please use consistent terminology.*

We have chosen the term "azimuth set" and modified it here and in section 5.1.

*296-298: I think this method of "manually checking the largest spacing values" should be elaborated. This "censoring by sand" cover could be illustrated in a figure, if it is not already, in which case reference the figure accordingly. This also seems like text that should be in the methodology section.*

In Figure 3a and b we have illustrated an area of sand and its impact on the spacing measurement, we have also indicated the area of maximum uncensored spacing. We have also moved part of this paragraph to the methodology section 3.1.2. (line 172-174)

*341: Method for connectivity calculation is not included in the methodology section. Add it there with appropriate citations. I also do not find the term "connectivity" to be suitable for the simple calculation of intersection nodes per square meter. I would suggest using another term such as "intersection node intensity" to avoid confusion with e.g. "Connections per trace/branch" from Sanderson and Nixon, 2015. If you wish to use "connectivity", please clarify the meaning of the term in this manuscript clearly in the methodology section.*

This is an accidental oversight on our part, so we have added the calculation of this parameter to the methodology section in *"General network parameters"* line (160-165). We have also changed the term connectivity according to your suggestion "*INi*" and removed the sentence from Figure 6 caption. (line 367)

*362-363: The classification of deformation structures should be included in the methodology section.*

We refer to the deformation structures glossary from Peacock et al. (2016) (see section 3.2.2, line 214-215).

*385: Please rephrase this to be clearer: "... but remain cataclastic deformation bands"*

We have rephrased as follows: *"e.g. deformation bands tend to be thicker in coarser sandstones and thinner in fine-grained sandstones but still show cataclastic deformation."* (line 398-399)

*433-435: The observation of fracture density correlating with permeability is observable from Figure 8d but I would not mind there being a separate figure where (mean) permeability for each well is plotted as a function P10 fracture density. If you have the interest, I recommend experimenting and seeing if such a plot would be clarifying for the readers.*

We believe that this could be part of a future study focusing on the petrophysical properties of the fractured Tchirezrine II reservoir, with more detailed characterisation of the various deformation structures and adequate characterisation of the petrophysical properties of the host rocks.

*457-458: Refer to the Figure from which these observations are made.*

As this dephasing can already be seen in the restitution curves in figure 9b, we have chosen not to add the piezometric curves over time to the data presented in figure 9.

*478: Specify what "Detailed analysis". E.g. "Orientation analysis of lineaments interpreted from satellite ...".*

Done (Line 489).

*479: Refer also to Figure 3?*

Done (Line 490).

*481: "... worth mentioning that length distribution of ...". Please rephrase. What do you mean by "length distribution" in this case?*

We have rephrased by *"It is worth mentioning that exponential distribution of lineament length of…"* (line 492-493)

*483: "... seems much more effective ...". Please rephrase. E.g. "... censoring bias has a greater effect on ...".*

Done (Line 495).

*485: Estimating the effect of truncation based solely on visual analysis of these log-log plots is not sufficient to rule out truncation ("... much lower than the main bend ..."). See my comment for Lines 171-172. You should note the caveats of your power-law fitting methodology somewhere here. The observations you have already made should be still be valid, especially when related to comparisons between sets.*

We have added a remark on the problems associated with the least-squares fitting method highlighted by Clauset et al. (2009). We also warn that the results must be interpreted parsimoniously and solely for the purposes of comparison between sets of lineaments. (Line 500-502)

*476-529: This subsection seems redundant. In here you list 4 main points. Then in the conclusion section you list 3 main points. I think trying to move all text from this subsection to later discussion subsections and to conclusions should remove the repetition between this redundant subsection and the conclusions. This subsection could also be renamed to simply "Limitations" and used to only discuss the method, data, etc. limitations of this study although these things can also be mentioned in the later subsections. You could also provide recommendations for future studies in such a section but in that case the subsection would be more fitting as the final one.*

This section is designed to provide an initial overview of the main results and their inherent limitations. This serves as a basis for the discussion as a whole and is not intended to give the main conclusions. We believe this section is necessary so that a reader unfamiliar with each of the methods employed can priorly benefit from them when interpreting the data.

*540-541: Please rephrase: ".. mechanical units .. are affected by these sets". Do you mean that the sets are contained by the mechanical units?*

We have rephrased as follows: *"…suggest that these sets are contained by mechanical units of several meters thickness."* (Line 554-555)

*558-560: Please rephrase the sentence starting with "By the way". I do not understand the Fossen citation here, "By the way" is not appropriate in a scientific text and what do you mean by "background deformation" in this context, please elaborate.*

We have changed "by the way" by *"Furthermore"*. The Fossen citation was refering to structure's terminology we have remove it, since we use glossary from Peacock et al. (2016). For background deformation, we refer to deformation structures related to the growth of fault systems. Mayolle et al. (2023) show that small-scale distributed structures link together over time, leading to the formation of master faults, localizing deformation, and abandoned structures as faults grow. We have reformulated this section to be clearer in the manuscript. (line 574-576)

*558-563: This section requires a bit more explanation. As I understand, you point out that discrete faults are often associated with secondary structures (mode I fractures) and due to the presence of these secondary structures the length distribution might follow an exponential curve. However, these secondary structures can be oriented differently in comparison to the main fault and subsequently, will not affect the set-wise length distribution. This should be mentioned.*

This is right, faults DZs show structures with various orientations, depending on the stress orientation-perturbation around the main fault. However, it is expected that secondary faults and wall-damage mode I fractures are preferentially subparallel to the main fault and constitute the dominant part of the damage zone. To be clearer on this point we add the end part of the sentence line 578-579.

*561: How is the presence of N°070E faults relevant in this context? The language makes it seem like the text continues from before "Also note that ..." when as I understand, this other set of faults is just a separate observation? You shouled try to introduce all the sets of faults first at the start of the paragraph on Line 545. Then go into the details.*

This is effectively an additional observation from the literature it shows that E-W deformation structures are described as related features of larger N070°E fault clusters observed across the basin and the basement. This observation is also relevant for the actual understanding of the structural framework of Imouraren deposit where large N070°E faults are observed mainly associated with smaller E-W deformation structure. So, we moved up this part of the paragraph on line 565-567.

*628-629: How does the clay infilling simultaneously seal in the E-W direction and also guide it? Does the clay form well-sealed channels within the fractures in the E-W direction? Please clarify shortly.*

The initial structure of the sentence is misleading, to avoid confusion, we have rephrased as follows: *"…clay infilling which probably forms E-W trending seals."* (Line 645)

*635-639: I think this could be removed as the tests are not present in this study.*

Although these tests are not presented in the study, their mention here allows us to temper the E-W anisotropy of the studied zone. The study of a single site (IMOU_2527_2 / 3 / 4) does not allow a generalization of the interpretation to the entire Imouraren deposit, and even less to the entire Tchirezrine II reservoir.

*647: Check the text inside parentheses.*

Done (Line 663)

*662-683: See comments on Lines 476-529. As I understand, the three conclusions 1, 2 and 3 come (mainly) from subsections 5.2, 5.3 and 5.4, respectively. This is great! However, the conclusion sections lacks reference to lineament analysis, partly due to different terminology. You use terms such as*

*"fracture" and "joint sets" instead of lineaments which make it seem like the lineament analysis has been forgotten in the conclusions.*

For this study, we use lineament analysis as a description of the general deformation network of the Tchirezrine II unit. This leads to various interpretations on the nature of the different sets of lineaments (see section 5.2), which are discussed in relation to other results and the literature. Since the general subject of this article is related to the characterization of sandstone NFR in the context of ISR from multiple methodologies, we not only consider the results as main conclusions, but try to highlight the interpretation built from them. This is also a reason why we choose not to merge the main results from section 5.1 and the conclusion.

**Figure and table comments**

*Figure 1:*

*1) Please include a legend for subfigure B. In the legend I would primarily like explanations of the black lines which can be assumed to be faults but what is the significance of the line width and the dashed line segments? The lithostratigraphy coloring from subfigure E is applicable to subfigure B, I assume. Please mention this in the subfigure B caption, it does not have to be repeated in a legend.*

We have added a legend for sub-figure (b). The black lines represent the interpretation of the faults from Orano's internal data. The wider lines correspond to the major faults in the basin and the dashed line is related to confidence in the location of the fault, as sand deposits make observations difficult.

*2) Is "Arlit F." fault segmented into two parallel traces in subfigure B? If not, how is "Arlit F." crosscut by the cross-section in subfigure C? Please label this segmentation in subfigure B if it is the case. I would also check if the cross-section is properly drawn on subfigure B. The faults in the west seem to be at the edge of the cross-section in subfigure B while in subfigure C they appear closer to the center. It is understandable that fault locations change as the detail/scale changes but the subfigures B and C do not seem to correspond to the same data currently or I am misreading the figure.*

At a lower scale there is many subparallel segments in the Arlit fault, the cross-section (c) is also from Orano's internal report and is based on exploration drills that why we can observe differences. We have checked for the location of the cross-section on map (b) and it appears that the western part should be longer, we also note that the horizontal scale in subfigure (c) is incorrect and should be 1500 m and not 500 m. We modified the caption and subfigure (b) and (c) to make better correspondence between the different map/cross-section. Thanks.

*3) Subfigure D has bad resolution. I would separate it entirely onto a new figure, add a legend, north arrow and more background detail, if available. This map of wells can be referenced often in the manuscript, so it should be well detailed. What are the black lines?*

We have slightly reduced subfigure (b) and enlarged subfigures (c) and (d). We have also added the Magagi fault names in subfigure (d), the same as one mentioned in figure 11.

*4) What is the geology based on? Please cite prior papers appropriately in the caption as has already been done in subfigure E caption for lithostratigraphy. If everything is based on this source, modify the caption to represent this.*

This map is modified from Orano internal database. (line 120-12)

*Figure 3:*

*1) The contents of subfigure B are almost impossible to discriminate e.g. in circle Z4 the "Censored lineaments" are very difficult to discriminate from "Lineaments". I suggest only showing the lineaments, no nodes or discrimination between censored and non-censored as these details are anyway not visible in the plots in the current form. The "fault segment" annotations should stay and they will also become better highlighted as other details are removed from the subfigure. This simplified version of subfigure B is suitable here but I would also then take the current versions of subfigure B, separate the individual target areas and include them in the Appendix in sufficient detail that e.g. nodes and censoring are visible.*

As suggested, we have made a simplified version of this subfigure (b) as well as a supplementary material (2) with better resolution.

*2) In subfigure C, please clarify what is "n" in this case (number of lineaments?).*

We have updated the caption (line 290-294).

*3) In subfigure A for Z2, the fault segments in the image are almost not visible. Please annotate them similarly to Z1 and Z4.*

Done

*Figure 6: The basis of this figure is not mentioned in the methodology section. What are the possible details that can be captured from the plot that are not already available from Table 2. Please include these details.*

We have added a description on the methodology section. We believe the representation of these data from Table 2 in the graphs of the figure 6 improved the description and comparison of the results. (Line 370-371)

*Figure 7: In subfigure D, could a well that intersects a fault be shown as one of the four wells and the fault in the log highlighted? The presence of faults in the wells would then be clearer to the reader instead of only being mentioned in text and imaged in subfigure C.*

In subfigure 7d, we have added a part of the IMOU_1471_2 log that shows the distribution of structures along the fault damage zones and the core zone. We have also added several reference to this subfigure in the manuscript.

*Figure 8: I find the double variables on Y-axis to be a bit distracting (NMR and Hydraulic conductivity, as I understand). I would examine if including both is necessary.*

As the topic of this article is related to both NFR and ISR communities (both geologists and hydrogeologists) we chose this option which is, in our opinion, more accessible for the different communities.

*Figure 9: 1) Is the figure missing a subfigure label C for this sentence: "Plan view of the estimated extension of the drawdown cone". Please rephrase/clarify. 2) I think an explanation of the bottom part of subfigure A in the caption is in order: What do the different axes of the ellipse represent? What does it mean that the other axis is longer than other?*

This plan view of the extension of the drawdown cone is a schematic representation of the 1/10 ratio on the extension of the drawdown. In the end, we believe that this representation is misleading because there are only two known orientations, and the shape of the ellipse could be significantly different. We have chosen to remove this part of sub-figure (a).

*Figure 11: It would preferential if you did not introduce new geological structures in a figure in the discussion section. E.g. I do not see reference to "Igagam F." in Figure 1? What is "IMOLA"? I again suggest making a separate version of Figure 1d where these details could probably be added (See comments on Figure 1).*

We have added Magagi F. in subfigure 1d and removed "Igagam F." from Figure 11a because this fault is not so relevant to understanding the conceptual model. IMOLA is the name of the southern part of the Imouraren deposit, we have removed it to not add complexity to the figure.

*Table 1: Explain abbreviations "nbr" and "Sd" in caption.*

Done (Line 296-297)

*Table 2: Caption contains text that should be in the methodology section. I suggest focusing on explaining just the figure in the caption and move the method explanation to the methodology section.*

As you suggest it, we have removed this part of the caption and moved it to the methodology section. (line 160-165)

**Responses to CC1: Giacomo Medici**

Dear Mr Medici,

We would like to thank you for your comments on the manuscript which, in addition to referees' comments, helped us to improve its quality. You will find detailed responses to your comments below.

Kind regards,

Maxime Jamet

**General comments**

*Interesting and original paper with interest in the fields of structural geology, hydrogeology, reservoir engineering and economic geology. Please, see my comments to improve the manuscript.*

**Specific comments**

*Line 19. Better just "fluvial context"? If there are some overbank mudstones they are anyway fluvial deposits.*

Done. (line 20)

*Lines 33-55. Insert recent review manuscripts that show similar hydro-mechanical models of fluvial sandstones with angle on water resources management and nuclear waste repositories.*

*- Review of groundwater flow and contaminant transport modelling approaches for the Sherwood Sandstone aquifer, UK; insights from analogous successions worldwide. Quarterly Journal of Engineering Geology and Hydrogeology, 55(4), https://doi.org/10.1144/qjegh2021-176.*

*- Reply to discussion on 'Review of groundwater flow and contaminant transport modelling approaches for the Sherwood Sandstone aquifer, UK; insights from analogous successions worldwide' by Medici and West (QJEGH, 55, qjegh2021-176). Quarterly Journal of Engineering Geology and Hydrogeology, 56(1), https://doi.org/10.1144/qjegh2022-097.*

In this part of the introduction, we don't believe it's appropriate to refer to modeling-oriented studies in order to be consistent with other references. Nevertheless, we'd like to thank you for introducing us to this study, which we have referenced in section 5.4. (line 672)

*Line 61. Please, provide more conceptual detail on the definition of Mode 1 fractures with regards to shear and relationship with other modes.*

Following a comment from other referees, the manuscript refers now to an "extensional fracture".

*Line 84. Clearly state the objectives of your research by using numbers (e.g., i, ii, and iii). The objectives should be three to fit your conclusions.*

This is done. (line 86)

*Lines 105-120. Add detail on the paleo-environments which are related to the lithotypes that you describe.*

For the purposes of this study, we believe there is sufficient information in this section to clearly understand the lithostratigraphic framework of the Tim Mersoi basin.

*Lines 121-123. Borehole images? ATV or OTV, both? Please, be more specific.*

As specified later in the manuscript, we used optical images. Also, this sentence is redundant with the end of the introduction, we have chosen to remove it.

*Line 122. "Geophysical logging", which techniques? If you disclose the techniques here is much easier for the reader.*

See previous answer.

*Line 167. Good to hear that you cite Noelle paper after many years!*

We're glad to hear it!

*Line 217. Did you back-up NMR porosity with the Archemede's method at least for some plugs from the cores? NMR porosity can provide issues with iron minerals which are present in fluvial sandstones. Errors can be up to 20%.*

We mention this issue in section 5.1 (iii), as far as we know, iron minerals shorten T2 relaxation time and minor derived porosity and permeability. Orano Mining has carried out an analysis of this potential problem on numerous plugs from two wells that are not part of the present study. They conclude that the porosity of the Imouraren sandstone facies does not diverge significantly from the Total NMR porosity (*TPOR*) measurements. However, as mentioned in this study, the estimated permeabilities in fractured boreholes are higher for equivalent facies. We believe that in the case of Imouraren, this is the dominant process influencing the NMR measurement. You mention errors of up to 20%. If you have any references on this subject, please let us know.

*Lines 456-464. I have noticed that your don't use the word "aquifer test". I suggest to use it.*

Done

*Lines 456-464. I leave to the author the possibility to extrapolate or not the value of aquifer transmissivity (T) / hydraulic conductivity (K). It takes no more than an afternoon. However, if this point is outside your objectives, you can leave it.*

This is indeed outside our scope for this study, but Orano's hydrogeologists have already taken the initiative to do so in order to best meet their industrial challenges.

*Lines 475-655. The proposed characterization has a clear link with flow modelling in either EPM or DFN scenarios. I suggest to mention / empathize this point in the discussion.*

As you rightly suggested, we've added a mention and references to flow modelling at the end of section 5.4 including Collet et al. (2022), Medici and West (2022) and Sharifzadeh et al. (2018). (line 669-674)

*Line 684. Add two or three sentences as final remarks after your three conclusive points.*

We believe that there is no specific need here to add any final remarks to the conclusion. However, if the editor requests it, we will do so.

*Lines 699-934. Please, integrate recent and relevant literature on sandstone aquifers suggested above.*

Done

**Figures and tables**

*Figures 4, 5, 6, 8, 9 and 10. I suggest to use the black colour for letters and numbers on the axes.*

Done

---

## Author Response (AR2)

**Responses to Nikolas Aleksi Ovaskainen's report.**

Dear Mr Ovaskainen,

once again, the authors thank you for your work as peer reviewer. Your comments in the interactive discussion section and in report #1 have greatly helped us to improve the quality of our manuscript.

We clarified the points you raised in the manuscript and also made corrections to ensure the consistent use of British English. In the following document, you will find a point-by-point response to your comments in Report #1.

We hope that these new corrections have consistently improved the manuscript and will enable this work to be published in Solid Earth.

Kind regards,

Maxime Jamet

**General comments related to prior review**

> *Furthermore, you should make sure you use either American or British English. Currently e.g. both "characterization" and "characterisation" are found in text.*

*This is still a problem. E.g. line 43 vs. 74.*

We have changed it as well as in line 674. We have also modified several words in the rest of the manuscript to better fit British English.

> *In terms of terminology, I think you should critically review terms related to fractures. You use e.g. "mode I fractures", "joints", "lineaments", "faults", "fractures" and "deformation structures" intermixed and sometimes in inappropriate context. Try to simplify the terminology (mode I fractures = joints?) and refer to the same features with the same terminology consistently. You should also consistently refer to azimuth sets, they are sometimes joint sets, sometimes fracture sets and sometimes orientation sets.*

*You have improved terminology. However, I still see some intermixed use of e.g. "orientation set" (Line 322). Please also examine of use of "joint set" is necessary on Line 685. Maybe "azimuth set" would be suitable?*

We have changed these terms in agreement with your comment (lines 326 and 687).

*Check Lines 24 and 393 for if "Mode I" can be replaced with "extensional open …" as you have done elsewhere.*

Thank you for bringing these oversights to our attention. We have corrected these terms (lines 24 and 396).

*Regarding your response to my previous referee report part on lines 245-262 ("We understand your point of view …"):*

*Thank you for the explanation but once again, I suggest letting any future readers also know your reasoning! You could add the info you provided me to e.g. section 3.1 (Lines 127-128). Explain, like you already did, why you did lineament interpretation in two scales and how the uses for the two scales differ and maybe, if suitable, why you chose not do the same analyses for both scales.*

We have added information about our choices and intentions for this part of the study (lines 129-132).

*Regarding your response to my previous referee report part on lines 662-683 ("For this study, we use …"):*

*I think some of this explanation would fit on Lines 487-488. Also examine if the 5.1 title could be changed to better clarify the purpose of the subsection. Otherwise I believe this subsection of "intermediate conclusions" will be confusing to readers, as it was for me. Furthermore, some of the explanation you provided in your response could probably be added to the start of the "Conclusions" section.*

The section 5.1 is now entitled "Results and limitations". We have specified that the results presented in this section are then discussed in the rest of the *Discussion* section (lines 491-492).

We have also modified the beginning of the conclusion by adding some of these explanations (lines 684-687).

**Specific comments by line(s):**

*15: "its impact" -> "their impact"*

Done (line 15).

*47: Please clarify in the text this part: "penalizing for petrophysical properties". E.g. "... properties by draining or compartmentalizing the reservoir", not only for me!*

We have clarified this in agreement with your comment (line 48).

*79: "... optical borehole imagery, geophysical data ..." Should there be "and"?*

Done (line 79).

*132: Please add info on what resolution the basic-scale orthomosaics are, if you have the information available.*

There is not only one basic-scale resolution for this orthomosaic since the used data are from three different satellite images. We cannot provide precise information on the initial-scale of the Landsat / Copernicus satellite since we do not know which Landsat generation is used by Google® to build this orthomosaic.

*136: "sampling area" -> "sampling areas"*

Done (line 140).

*147: "... of lineament networks ..." -> "... of lineament (and fracture) networks ...". This better covers the themes of the cited publications.*

We agree with your comment and have added your proposal (line 151).

*160: "fracture set" is a bit misleading here. I suggest "To describe the quantity of fractures, Dershowitz ..."*

We have modified it (line 164).

*162: "calculates" -> "calculated". I suggest: "which allow to better ..." -> "... which allows the better comparison of their spatial distribution".*

We have changed this sentence accordingly to your suggestion (line 166).

*180: "lineaments set" -> "lineament set"*

Done (line 184).

*193-195: You explained your reasoning well in your response but I would prefer it if, at least partly, the reasoning is included in the text to avoid the readers getting confused on where the cut-off comes form as I was before your explanation. Currently, I find the text "in accordance with other studies ... and the resolution of satellite images" to not explain it well enough.*

We have rephrased this part of the paragraph to better explain our reasoning: "Considering the resolution (0.3 m/pixel) of the images data, we have chosen a 6-meters value for truncation cut-off in accordance with other studies (e.g., Bonnet et al., 2001; Soliva and Schultz, 2008). If we consider a reasonable margin of error of 2 pixels (0.6 m) when sampling the lineaments, the length values are not truncated greater than 1/10 of their real length and this ratio decreases as the length of the lineaments increases." (lines 197-200).

*205-206: Be specific or introduce as examples. Assuming other deformation structures are mapped: "... structures such as fractures and deformation bands."*

Done (line 212).

*393: "factures" -> "fractures"*

Done (line 396).

*517: "We however have not enough data to precise these effects" -> Please rephrase. E.g. "We however do not have enough data ...".*

Done (line 520).

*603: Should it be: "relevant to analyse any correlation ..." -> "relevant to analyse regarding any correlation ..."*

True (line 606).

*645: "form" -> "forms"*

Done (line 648).

*690: "host" -> "hosted"*

Done (line 693).

**Figure and table comments**

*Figure 1d: "Suposed" -> "Supposed"*

Done.